# Large-scale phenotyping and comparative genomics reveal genetic features of *Listeria* persistence in epithelial cells

Aurélie Lotoux[1], Matthieu Bertrand[1], Pierre-Emmanuel Douarre[2], Mounia Kortebi[1], Hélène Riveiro[1], Federica Palma[2], Goran Lakisic[1], Edward M. Fox[3], Laurent Guillier[2], Anna Oevermann[4], Sophie Roussel[2], Hélène Bierne[1], Alessandro Pagliuso[1]*, Eliane Milohanic[1]*

**1** Université Paris-Saclay, INRAE, AgroParisTech, Micalis Institute, EPIMIC Lab, Jouy-en-Josas, France, **2** Maisons-Alfort Laboratory for Food Safety, Salmonella and Listeria Unit, University of Paris-Est, French Agency for Food, Environmental and Occupational Health & Safety (ANSES), Maisons-Alfort, France, **3** Department of Applied Sciences, Northumbria University, Newcastle upon Tyne, United Kingdom, **4** Division of Neurological Sciences, Department of Clinical Research and Veterinary Public Health, Vetsuisse Faculty, University of Bern, Bern, Switzerland.

☉ These authors contributed equally to this work.
* alessandro.pagliuso@inrae.fr (AP); eliane.milohanic@inrae.fr (EM)

## Abstract

During infection in epithelial cells, after invading the cytosol, multiplying, and spreading, *Listeria monocytogenes* (*Lm*) ceases to produce ActA and becomes trapped in *Listeria*-containing vacuoles (LisCVs). These persistence acidic vacuoles harbor bacterial subpopulations that resist to stress in a metabolically dormant state. Although LisCVs have been proposed as a hallmark of *Lm* persistence in epithelial cells, their prevalence across strains and the bacterial factors underlying their formation remain uncharacterized. Given the significant genetic diversity within the species, it is important to consider this variability when studying persistence phenotype. Therefore, we screened over one hundred *Lm* isolates spanning two major evolutionary lineages and belonging to 23 clonal complexes from diverse ecological origins. Strikingly, the vast majority of strains, including both clinical and environmental isolates, were capable of forming LisCVs, suggesting that vacuolar persistence is a widespread and conserved feature of *Lm* pathogenesis. Nevertheless, among the group of hypo-virulent strains mostly associated with food and carrying a truncated InlA, we identified four isolates with an altered persistence phenotype. Two of them showed defects in the early stages of infection and carried mutations in key virulence genes (*hly* and *gshF*). The other two, instead, were specifically affected in the persistence stage by showing a reduced ability to form LisCVs. Comparative genomic analysis revealed that a mutation in the *folP* gene, required for folate biosynthesis, was responsible for impaired persistence. Live-imaging and microscopy analysis highlighted a reduced bacterial motility and intercellular spreading of the *folP* mutant, although the level of ActA at the bacterial surface was increased. Together, our work identifies folate

**Data availability statement:** All data underlying the findings of this study are fully available without restriction and all genomic datasets used in this study are listed in S2 Table.

**Funding:** This work was supported by grants from the Agence Nationale de la Recherche (ANR, https://anr.fr) to H.B. and E.M. (PERMALI: ANR-20-CE35-0001-01) and to A.P. (THOR: ANR-20-CE15-0008-01); from the Université Paris-Saclay (https://www.universite-paris-saclay.fr) to H.B. (DEPISTALIS, AAP Poc in labs 2019) and A.L.; from INRAE's MICA department and Micalis Institute (https://www.inrae.fr) to E.M. and A.P. This work was also supported as part of France 2030 program "ANR-11-IDEX-0003". The funders had no role in study design, data collection and analysis, decision to publish, or preparation of the manuscript. M.B. received a salary from the Agence Nationale de la Recherche (ANR).

**Competing interests:** The authors have declared that no competing interests exist.

biosynthesis as a critical metabolic pathway governing *Lm* persistence by regulating ActA levels and activity. Downregulation of ActA at the bacterial surface is therefore a crucial event for the establishment of the intracellular persistent niche during long-term infection of epithelial cells.

## Author summary

Long regarded as a strictly cytosolic pathogen, *Listeria monocytogenes* is now revealing a dual lifestyle that includes intracellular vacuoles. Recent evidence shows that in epithelial cells this bacterium can enter a dormant state within acidic vacuoles, termed LisCVs, potentially contributing to silent carriage and antibiotic treatment failure. Here, we reveal that vacuolar persistence is a widespread and conserved feature among diverse *Listeria* strains. Strikingly, we identify folate metabolism as a key regulator of this phenotypic switch, linking metabolic cues to the shutdown of bacterial motility. Our findings uncover a novel connection between central metabolism and intracellular niche adaptation, shedding new light on how *Listeria* survives and hides within the host.

## Introduction

*Listeria monocytogenes* (*Lm*) is a widespread facultative pathogen that causes listeriosis, a life-threatening foodborne disease that often includes sepsis, meningoencephalitis and miscarriage [1,2]. Although *Lm* is a common contaminant of various food products and is widespread in agri-food environments, listeriosis remains rare, with an average of 300 cases per year in France [3]. This suggests that exposure to *Lm* rarely induces clinically relevant symptoms, but instead results in silent colonization. In support of this, a significant asymptomatic fecal carriage of *Lm* in healthy humans (~10%) and animals has been reported [4,5]. Moreover, the presence of *Lm* in wildlife, particularly in wild mammals has also been documented showing that diverse species can be vehicles for this pathogen [6,7]. Overall, the asymptomatic carriage of *Lm* in humans and farm/wild animals might represent an important outcome of *Lm* infection, yet the molecular mechanisms at play remain poorly explored.

Upon infection, *Lm* deploys a variety of virulence effectors to target different cellular components and therefore subvert host cell activities. In non-phagocytic cells, *Lm* induces its internalization via internalin proteins InlA and InlB, and then disrupts the vacuolar membrane using the pore-forming toxin listeriolysin O (LLO). Once in the cytosol, *Lm* hijacks the host actin machinery using the motility factor ActA, that propels the bacterium intracellularly and facilitates spread to neighboring cells, thereby promoting bacterial dissemination. Although *Lm* has been considered for a long time as a cytosolic pathogen, there is a growing body of evidence that this bacterium can also inhabit intracellular vacuoles [8–12]. In macrophages, *Lm* can reside in spacious Listeria-containing phagosomes

(SLAPs) that support slow intravacuolar growth under conditions of limited listeriolysin O activity [8]. Recently, by using fluorescent reporters in epithelial LoVo cells, a fraction of Lm was seen to remain for hours and even multiply within neutral vacuoles (eSLAPs, for epithelial SLAPs) positive for Rab7, LC3 and LAMP1 [10]. These vacuolar bacteria later escaped to the cytosol, indicating a transient intravacuolar replication niche. In bovine neutrophils, phagocytosed Lm often fails to escape to the cytosol and persists within single-membrane vacuoles [13]. Many of these intravacuolar bacteria enter a VBNC state yet can survive and spread to other cells, suggesting that neutrophils may act as mobile vacuolar niches during infection [13].

We have recently shown that during long term infection in epithelial cells, after the active dissemination phase, bacteria stop producing ActA protein on their surface and are engulfed in LAMP1-positive acidic vacuoles called LisCVs. This phase is accompanied by a phenotypic switch with bacteria entering a dormant/slow-replicative phase. Importantly, bacteria retain the capability to exit from this quiescent phase and return to the active proliferation and dissemination upon host cells sub-culturing in vitro. These observations suggest that LisCVs might represent an intracellular niche for quiescent Lm within tissues, which would promote Lm asymptomatic carriage [14,15]. Such persister stages might be epidemiologically important in potentially facilitating spreading and survival of Lm, particularly in relapse infections. It is therefore important to identify bacteria and host features that regulate LisCVs biogenesis to fully understand Lm-host interaction. Given the considerable genetic diversity within the species, it is also important to investigate potential strain heterogeneity in the ability to form and maintain these persistent intracellular forms.

Lm is divided into four main genetic lineages, several serovars [16–19] and more than 200 clonal complexes (CCs) [20,21]. Isolates belonging to lineage I, in particular CC1, CC2, CC4 and CC6 (serovars 1/2b and 4b) are strongly associated with clinical cases and under-represented in food. Conversely, lineage II strains mostly CC9 and CC121 (serovars 1/2a and 1/2c) are more frequently isolated from food samples and cause infections in highly immunocompromised patients. Therefore, CC1, CC2, CC4 and CC6 have been considered hyper-virulent, while the food-associated CC9, CC121 are mostly classified as hypo-virulent strains.

Genome analysis of hyper-virulent Lm clones has revealed the acquisition of accessory virulence-associated genes. These include the LIPI-4 locus, associated with neural and placental tropism in CC4 isolates; LIPI-3, which encodes listeriolysin S and facilitates gastrointestinal colonization; and the gltA-gltB cassette responsible for serotype 4b-specific glycosylation of the teichoic acids [21–25]. In addition to these accessory elements, allelic variations in sigB, encoding the alternative sigma factor B, have also been shown to modulate stress response and virulence [26].

Hypo-virulence is typically associated with virulence-attenuating mutations, particularly in the inlA gene, where premature stop codons (PMSCs) result in truncated InlA proteins, impairing host cell invasion. Such mutations are commonly observed in most food isolates of CC9 and CC121, and are linked to decreased epithelial cell invasion in vitro as well as reduced virulence in a guinea pig oral infection model [27–29]. In addition to a truncated InlA, mutations in prfA, plcA and hly are found among low-virulence strains and non-clinical isolates, but are distributed to diverse phylogenetically clades [30–34]. Together, this genetic and epidemiological heterogeneity of the population reflects variations in pathogenic potential among Lm isolates, which are important to consider.

In this study, we aimed at identifying bacterial determinants involved in vacuolar persistence during long-term infection in trophoblast JEG-3 cells. By combining a microscopy-based screening with comparative pangenome analysis, we identified Lm variants with altered persistence capacity. We report a role for the folate pathway as a mediator of bacterial persistence via its modulation on ActA protein levels and activity. Our study supports the hypothesis that genes involved in the intracellular lifestyle of Lm, including persistence inside LisCVs, might undergo a purifying selection in hypo-virulent strains. Food associated strains thus represent a better reservoir to look for genes involved in bacterial persistence.

## Results

### The majority of *L. monocytogenes* strains persist in epithelial cells inside LisCVs

We analysed the ability of *Lm* to persist in epithelial cells by screening 70 bacterial isolates for LisCVs formation in placental JEG-3 cells, a well-characterized epithelial model to study *Lm* persistence [11] (Fig 1A). In JEG-3 cells, *Lm* is internalized through the InlA- and InlB-dependent pathways [35] and form LisCVs after three days of infection [11]. We selected bacterial isolates of lineage I and II, the two major lineages of *Lm* [20,36], including the reference strains EGDe, 10403S, and LO28, a strain producing a truncated InlA (InlA-Δ) [37–40] (Fig 1A). Bacterial isolates were chosen to represent the ecological and genetic diversity of the *Lm* species (Fig 1A and S1 Table): 15 isolates from listeriosis cases (4 human and 11 animal isolates), 36 food isolates, and 17 environmental isolates (13 from animal farms and 4 from food processing environments (FPE)) (Fig 1A). In terms of genetic distribution, 32 isolates belonged to lineage I, including 19 of the four predominant clonal complexes (CC) found in human clinical listeriosis cases (CC1, CC2, CC4, and CC6), and 38 isolates belonged to lineage II, including 13 of the two predominant CC in European food isolates (CC9 and CC121) (Fig 1A) [20,36,41].

JEG-3 cells were infected and intracellular bacterial loads were determined at 2 h (entry) and 72 h post-infection (p.i.) (persistence) by enumeration of intracellular CFU count on BHI agar plates. After 2 h p.i., the internalization efficiency was estimated by the percentage of entry relative to the inoculum, with strain 10403S chosen as reference (i.e., % entry = 100%) (Figs 1B and S1A). Strains showed a large variability in the efficiency of entry, particularly with a lineage-dependent effect. Indeed, 81% (26/32) of lineage I strains, and 29% (11/38) of lineage II strains, had a higher percentage of entry at least two times higher than 10403S (Figs 1B and S1A). In addition, 37% (14/38) of lineage II strains and only 6% (2/32) of lineage I strains were hypo-invasive, as exemplified by the LO28 strain (Figs 1B and S1A). All CC121 and CC9 strains, except EGDe, were hypo-invasive, which is consistent with the fact that most of these strains have a truncation in InlA (Fig 1A and S1 Table) [42,43].

Despite this variability in entry efficiencies, the vast majority of strains (69/70) were able to multiply and persist over 3 days of infection, as shown by the quantification of intracellular bacterial loads at 72 h p.i. (Figs 1C and S1B). Notably, most of the hypo-invasive strains, including LO28, showed no differences in intracellular bacterial load at 72 h p.i., suggesting that even in the absence of a functional InlA, these strains retain their ability to proliferate intracellularly after internalization. For some strains of lineage I, we observed that infection could lead to significant cytotoxicity and cell damage, probably due to higher bacterial load or production of the bacterial toxin LLO, suggesting that these strains possess a higher virulence potential, which may not be accurately reflected by CFU quantification.

We next investigated the ability of *Lm* to localize to LisCVs by selecting a representative panel of 35 strains from different CC and origins. As a control, we used the three laboratory strains EGDe, 10403S and LO28. All strains, including InlA-Δ, formed LisCVs without any significant difference at 72 h p.i. as revealed by co-localization of *Lm* with the lysosomal marker LAMP1 (Fig 1D).

Together, these results showed that the majority of *Lm* strains have the capability to persist inside the LisCVs regardless their origin or clonal complex. We conclude that, a persistent phase inside vacuoles is a general feature of the intracellular lifestyle of *Lm*.

While most strains multiplied like the reference strains, one strain, 2965 (a CC121 food isolate), showed 100-fold reduction in bacterial load at 72 h p.i. (Figs 1C and S1B). 2965 was also found to be hypo-invasive (Fig 1B), most likely because of a truncated InlA protein (InlA-Q492) (S1 Table). We first confirmed that 2965 did not have an impaired growth by comparing its growth in BHI medium with that of the EGDe reference strain and 17SEL410LM (CC121-WT), a CC121 strain that behaves like EGDe during intracellular infection (Figs 1D, 1E, S1B and S2A). Thus, the decreased bacterial load at 72 h of 2965 (hereinafter referred to as variant V1) might be due to the acquisition of one or more mutations that specifically impair intracellular multiplication.

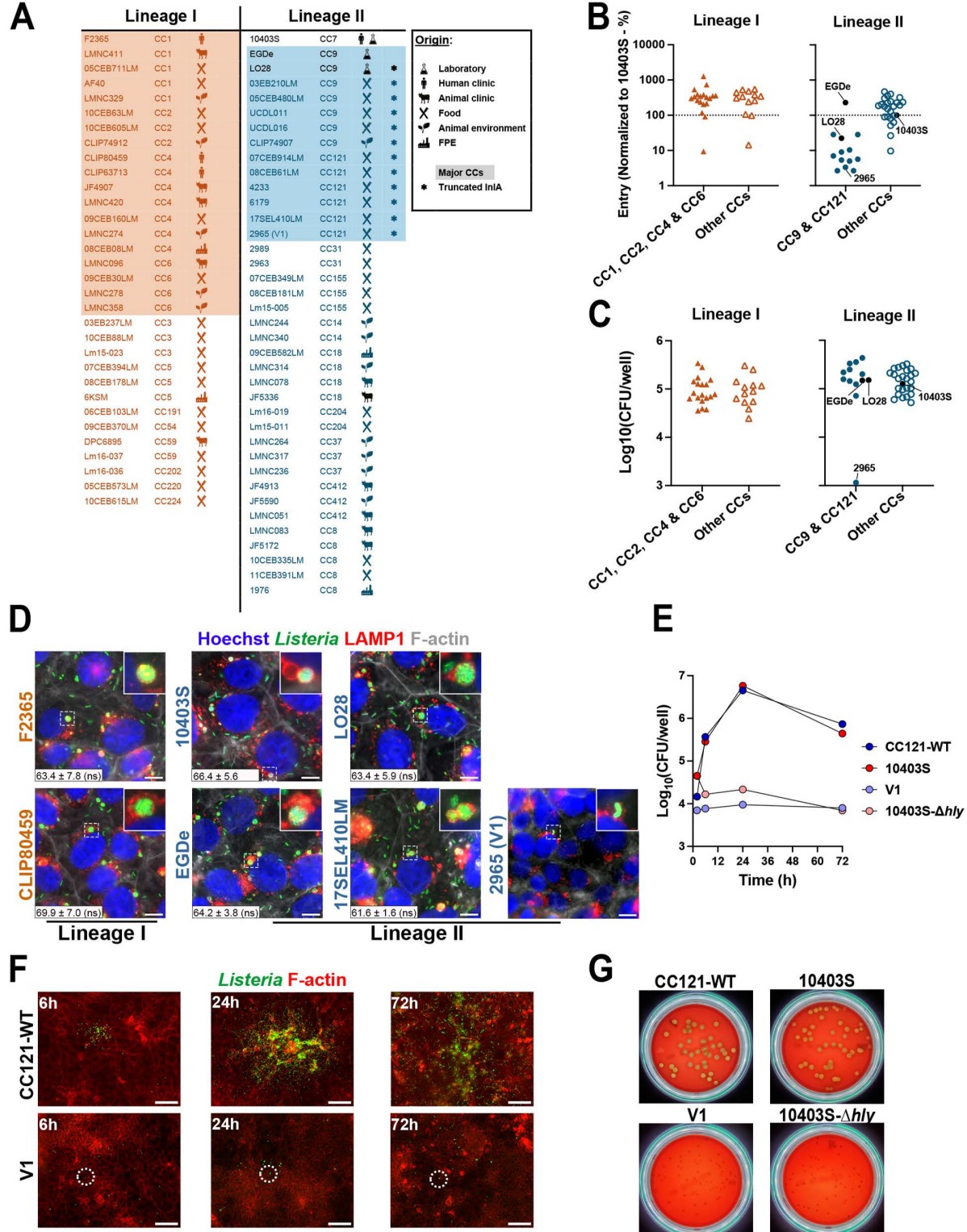

**Fig 1. Identification of bacterial strains with an altered persistence in placental JEG-3 cells. A**. Origin and characteristics of the 70 *Lm* strains included in the screen. **B.** Entry capacity of the strains assessed after 2 h of infection. The percentage of entry (calculated as the number of internalized bacteria divided by the number of bacteria initially added to the wells) was normalized to 10403S and represented by lineages and clonal complexes.

Entry values for laboratory strains 10403S, EGDe, and LO28 are indicated by tags and black circles. Data represent the mean of three independent experiments. **C**. Intracellular multiplication capacity assessed by quantifying the bacterial load at 72 h p.i. Results represent the mean of three independent experiments. The V1 variant (2965) is highlighted with a tag (see **B, C**). **D**. Representative micrographs of LisCVs in JEG-3 cells infected at 72 h p.i with strains 10403S, EGDe, and representative lineages I and II isolates, including the lineage II variant V1. Images show overlays of Hoechst (DNA, blue), *Lm* (green), LAMP1 (red) and F-actin (white) signals. Bars: 10μm. Insets show high-magnification views of the boxed regions with representative LisCVs. For each strain, bacterial association with LAMP1 is shown as a percentage at the bottom of each image and compared to that of the 10403S strain. **E**. Kinetic of intracellular growth for the CC121-WT control strain (17SEL410LM), the V1 variant (2965), the 10403S strain, and its isogenic mutant 10403S-Δ*hly*. **F**. Low magnification micrographs of JEG-3 cells infected with the CC121-WT control strain (17SEL410LM) and V1 variant at 6 h (left), 24 h (middle) and 72 h (right) p.i. Images show overlays of *Lm* (green) and F-actin (red) signals. Circles highlight individual bacteria. Bars: 50μm. **G**. Non-hemolytic phenotype of the *Lm* V1 variant assessed on horse blood agar after 24 h of growth. The CC121-WT and 10403S strains were used as positive controls, while the hemolysin-deficient strain 10403S-Δ*hly* served as the negative control.

We investigated whether V1 presented mutations in the virulence genes of the LIPI-1 locus, which are involved in key steps of the *Lm* intracellular life (i.e*.,* exit from the entry vacuole, actin-dependent motility and cell to cell spread) (Table 1). By analyzing the virulome, we identified a mutation in the *hly* gene coding for LLO, the cytolysin essential for exit of bacteria from the internalization vacuole [44]. A single nucleotide insertion at position 1383 induces a frame-shift and the generation of a premature stop codon leading to the production of a truncated LLO-Δ472 protein (Table 1). The loss of the C-terminal region, which is involved in the binding of this pore-forming toxin to cholesterol in eukaryotic membranes, inhibits the function of LLO (S2B Fig) [45,46]. Accordingly, V1 phenocopied the 10403S-Δ*hly* mutant, which we previously showed to be severely attenuated from the early stages of JEG-3 epithelial cell infection [11] (Fig 1E). At 2 h p.i., V1 and 10403S-Δ*hly* entered JEG-3 cells as efficiently as the control strains 17SEL410LM (CC121-WT) and 10403S, respectively, but stopped to multiply immediately after entry (Fig 1E). Like a LLO-negative mutant, V1 failed to escape the entry vacuole, multiply, spread in the cell monolayer and did not get trapped in LisCVs at 72 h p.i., unlike control strains (Fig 1D-1F) [11]. Additionally, V1 was non-hemolytic on blood agar plates, as compared with two hemolytic strains (10403S and the control strains 17SEL410LM (CC121-WT)) and the hemolysin-deficient strain 10403S-Δ*hly* (Fig 1G).

In conclusion, a screening of 70 strains for their ability to infect and persist in JEG-3 epithelial cells led to the identification of one hypo-virulent variant which, in addition to an *inlA* mutation, has acquired a mutation in *hly* leading to a dysfunctional LLO that prevented multiplication and persistence.

## Identification of three *Lm* InlA-Δ variants showing altered LisCVs formation

The identification of one strain with impaired bacterial load at 72 h p.i. among the InlA-Δ CC, lead us to hypothesize that the loss of functional InlA might counter-select genes required for bacterial intracellular life–such as those lacking LLO function– while also enabling the identification of strains specifically defective at later stages of infection, including long-term persistence inside LisCVs. We thus performed a second screening on 35 new InlA-Δ isolates and monitored both CFU and LisCVs formation at 72 h p.i. We included in the screening 25 CC9 (comprising LO28, EGDe and its isogenic mutant EGDe-Δ*inlA*) and 14 CC121 (comprising CC121-WT) (Fig 2A and S2 Table). We adjusted the MOI for each strain to obtain comparable number of bacterial entry events (20–100 intracellular bacteria per well at 2 h p.i.), which enabled analysis of bacterial replication and individual infection foci at 72 h p.i.

**Table 1. Comparison of the LIPI-1 virulence locus between EGDe strain, CC121-WT (17SEL410LM) and V1 (2965).**

| Genome | LOCI | CC | *prfA* | PrfA | *plcA* | PlcA | *hly* | LLO | *mpl* | Mpl | *actA* | ActA | *plcB* | PlcB |
|---|---|---|---|---|---|---|---|---|---|---|---|---|---|---|
| EGDe | 57 | CC9 | prfA_16° | 237 | plcA_1 | 317 | hly_1 | 529 | mpl_1 | 510 | actA_1 | 639 | plcB_1 | 289 |
| CC121-WT | 55 | CC121 | prfA_8 | 237 | plcA_18 | 317 | hly_13 | 529 | mpl_17 | 510 | actA_121 | 604 | plcB_9 | 289 |
| V1 | 55 | CC121 | prfA_8 | 237 | plcA_18 | 317 | **hly_13°** | **472** | mpl_17 | 510 | actA_121 | 604 | plcB_9 | 289 |

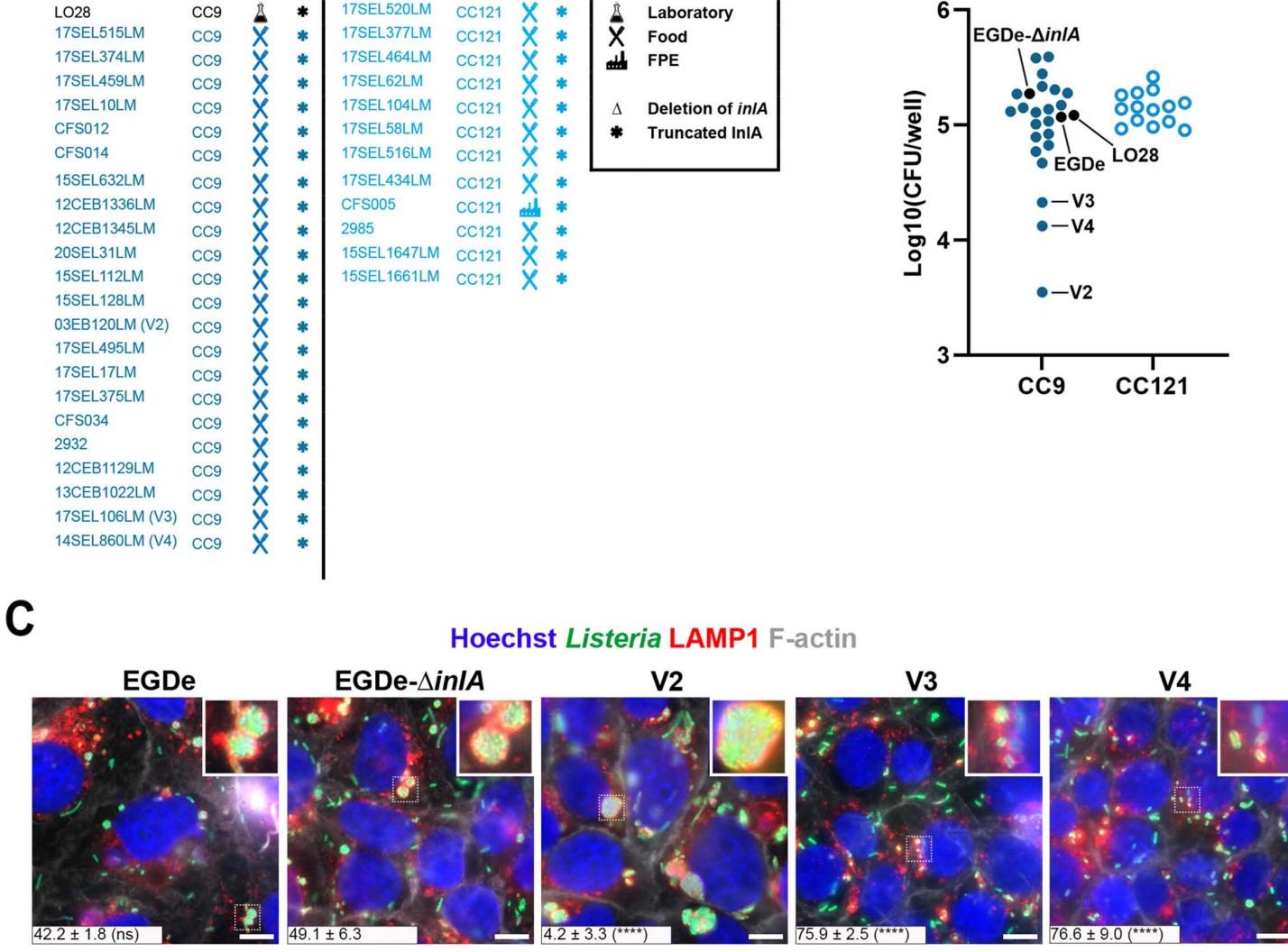

**Fig 2. Identification of three *Lm* InlA-Δ variants showing altered dissemination and LisCVs formation. A**. Origin and characteristics of the 39 InlA-Δ strains (25 CC9 and 14 CC121), including the reference strains EGDe, EGDe-Δ*inlA* and LO28 as well as the CC121-WT control strain (17SEL410LM). **B**. Intracellular multiplication capacity of the strains assessed by quantification of the bacterial load at 72 h p.i. in JEG-3 cells. Data are presented by clonal complexes. The three reference strains, as well as variants V2 (03EB120LM), V3 (17SEL106LM) and V4 (14SEL860LM) are highlighted with tags. The CFUs/well represent the mean of three independent experiments. MOIs were set to 0.001-0.05 for hypo-invasive CC9 and CC121 (InlA-Δ) strains and for the EGDe-Δ*inlA* mutant, and to 0.0001–0.0005 for control EGDe (InlA-WT) strain. **C**. High magnification micrographs of JEG-3 cells infected for 72 h with EGDe, EGDe-Δ*inlA* and the three variants V2, V3 and V4. Images show overlays of Hoechst (blue), *Lm* (green), LAMP1 (red) and F-actin (white) signals. Bars: 10µm. Insets show high-magnification views of the boxed regions with representative LisCVs. For each strain, actin association of LAMP1-negative bacteria is shown as a percentage at the bottom of each image and compared with that of the EGDe-Δ*inlA* strain.

Strikingly, we identified three strains showing a decreased bacterial load at 72 h p.i.: 03EB120LM, 17SEL106LM and 14SEL860LM (hereafter referred to as variant V2, V3 and V4, respectively) (Fig 2B). This finding suggests that either InlA-Δ isolates constitute a suitable source to find genes involved in intracellular infection or that the CC9 genetic background, to

which all identified variants belong, harbors specific factors that affect bacterial persistence. We next looked at their capability to persist in LisCVs by analyzing bacterial co-localization with the lysosomal marker LAMP1. Interestingly, while EGDe and EGDe-Δ*inlA* formed perinuclear LAMP1 positive LisCVs, V2, V3 and V4 presented altered phenotypes: V2 showed larger vacuoles than the reference strains, while V3 and V4 had an impaired LisCVs formation (Fig 2C). We attempted to quantify the proportion of LAMP1-positive bacteria for each variant, however, this proved technically challenging. Indeed, infections with V2, V3, and V4 were associated with very low cytotoxicity, resulting in highly confluent cell monolayers characterized by dense and continuous LAMP1 staining. Although discrete LAMP1-positive LisCVs could be readily identified, these conditions precluded reliable automated segmentation and global colocalization analysis. We therefore adopted a complementary approach by quantifying the proportion of actin-positive bacteria, which correspond to bacteria that have escaped LisCV targeting [11]. This analysis revealed that V2 displayed reduced actin association, consistent with increased retention within LisCVs, whereas V3 and V4 exhibited increased actin association and consequently reduced LAMP1 co-localization (Fig 2C).

To our knowledge this is the first identification of *Lm* isolates showing an alteration in the intracellular persistence. We therefore sought to identify the genetic determinants in V2, V3 and V4 responsible for these variations.

## V2 shows a "hyper-vacuolar" phenotype due to a deletion of the *gshF* locus

We first quantified the size of LisCVs formed by V2 at 72 h p.i. For this analysis, we selected discrete and clearly identifiable LisCVs. The LisCVs generated by V2 appeared larger (Fig 3A) and contained more bacteria (Fig 3B) than those of the EGDe-Δ*inlA* reference strain. This phenotype was reminiscent of that of the Δ*actA* mutant that is captured inside LisCVs more efficiently and also forms larger perinuclear LisCVs (Fig 3A and 3B) [11]. In addition, like the Δ*actA* mutant, the intracellular bacterial load of V2 measured by CFU started to decline as early as 6 h p.i., in contrast to EGDe-Δ*inlA* (Fig 3C).

These results led us to test whether V2 might have some defect in ActA expression and thus in intracellular spreading. We first assessed the ActA localization at the bacterial surface. Fluorescence microscopy revealed that intracellular V2 bacteria exhibited a significantly reduced expression of ActA at their surface compared to EGDe-Δ*inlA* (Fig 3D and 3E). Consistently, microscopy observation of infected cells revealed that V2 was strongly impaired in cell to cell spread at the very early stage of infection (S3 Fig). Furthermore, at 24 h p.i. V2 formed foci of dissemination significantly smaller than those of EGDe-Δ*inlA* (Fig 3F and 3G). We conclude that the hyper-vacuolar phenotype of V2 was likely the result of an impaired expression of ActA. We next sought to determine the mechanism responsible for ActA downregulation at the bacterial surface. Lower expression of ActA might depend either on mutations in the protein sequence that trigger degradation, or mutations in PrfA, the master regulator of ActA expression [47]. Genomic analysis however, showed that neither of these explanations was valid, as the LIPI-1 locus in V2 was shown to be identical to that of the WT EGDe strain (S3 Table). We thus employed a comparative genomic approach to identify V2-specific genetic features responsible for the spreading defect and LisCV size alterations. We selected EGDe and 19 CC9 strains that behaved as the reference strains (EGDe and EGDe-Δ*inlA*) in the second screening (Fig 2A and 2B). Of note, these strains were also tested for actin polymerization (6 h p.i.) and LisCV phenotype (72 h p.i.) (S4 Fig), showing no major differences compared to the reference strains. We used a software tool (Panaroo) [48] to detect the absence/presence of genes. The pangenome analysis revealed the absence of 24 genes, grouped into 3 distinct regions on the EGDe genome (Table 2): deletion 1, spanning from *lmo0082* to *lmo0094*, includes an operon encoding proteins of unknown function as well as an ATP synthase-like operon; deletion 2 comprises *lmo1655* and *lmo1656*, the latter encoding a virulence factor not expressed *in vitro* [49]; deletion 3, from *lmo2767* to *lmo2775*, encompasses a region including *gshF (lmo2770)*, which encodes a glutathione synthase required for PrfA activation [50] (S5 Fig). The 12 kb region missing in V2 was replaced by the insertion sequence IS1542, a transposase that likely mediated the loss of the 9 genes (S5 Fig).

The glutathione synthesized by GshF is involved in the late activation of PrfA in the intracellular environment, which in turn activates several genes involved in bacterial dissemination, including *actA* [50,51]. We thus reasoned that the decreased expression of ActA at the bacterial surface in V2 might be dependent on an impaired PrfA activation. In agreement with this hypothesis, re-expression of *gshF* in V2 via chromosomal integration fully restored

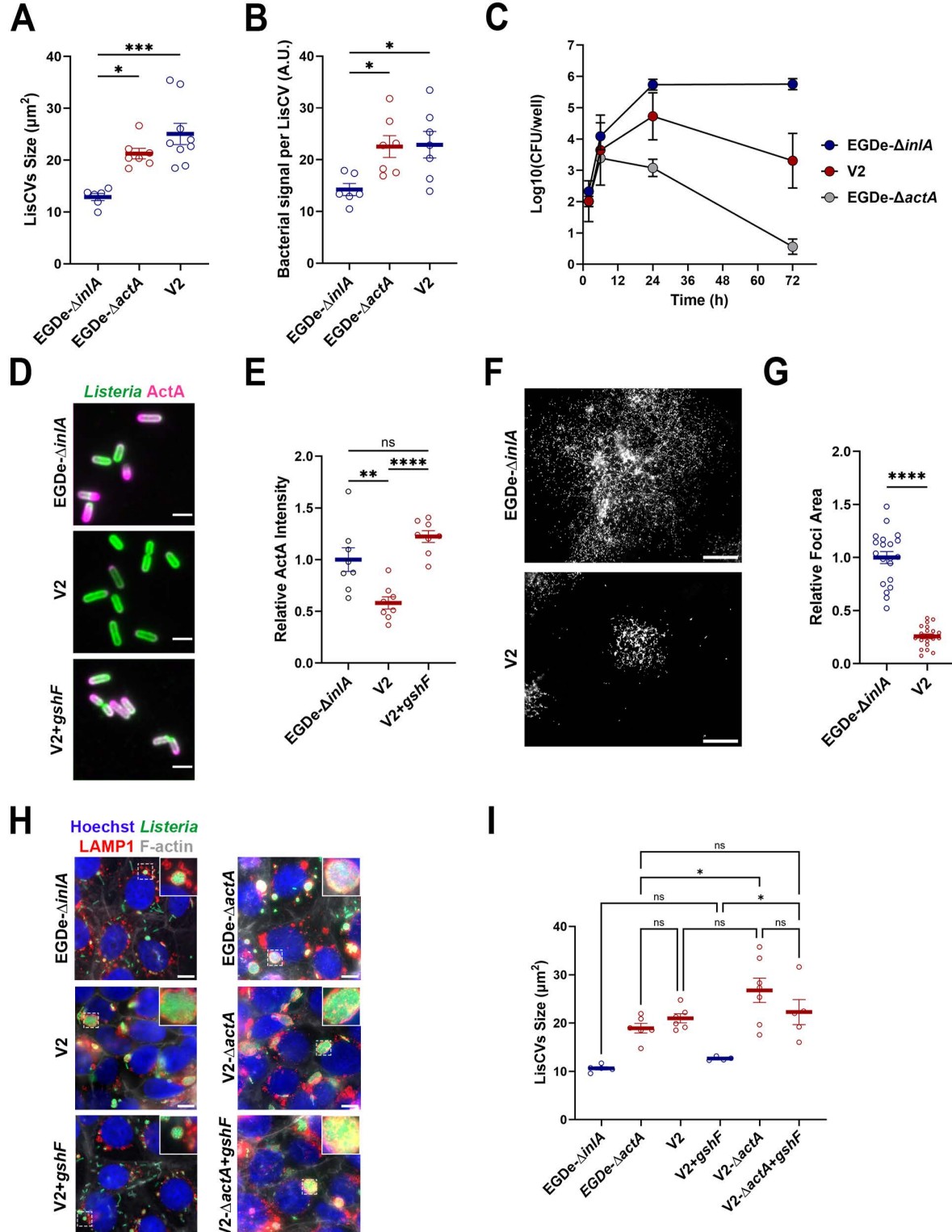

**Fig 3. The intracellular "hyper-vacuolar" phenotype of the V2 is associated with the loss of the *gshF* gene. A.** Histogram showing the size of LisCVs at 72 h p.i. in JEG-3 cells infected with EGDe-Δ*inlA*, EGDe-Δ*actA* and V2 (MOI = 0,01). Each point represents the average size of the LisCVs within a single microscopic field. Bars indicate the mean ± standard error of the mean (SEM) of 6 to 9 microscopic fields per condition, pooled from two

independent experiments. **B**. Average bacterial signal per LisCV (arbitrary units, A.U.) at 72 h p.i. in JEG-3 cells infected with EGDe-ΔinlA, EGDe-ΔactA and V2 (MOI = 0,01). Each point corresponds to the average bacterial signal per LisCV within a single microscopic field. Bars represent the mean ± SEM of 6 to 9 microscopic fields per condition, pooled from two independent experiments. **C**. Intracellular infection kinetics of variant V2 in JEG-3 cells (MOI = 0.01), compared to EGDe-ΔinlA strain (MOI = 0.01) and EGDe-ΔactA (MOI = 0,001) strains. Each point corresponds to the mean CFU count from two wells. Data shown are representative of three independent experiments. **D**. Representative micrographs of EGDe-ΔinlA, V2 and V2 + gshF complemented strain at 6 h p.i. in JEG-3 cells. Images show overlay of Lm (green) and ActA (magenta). Bars: 2 µm. **E**. Histogram of relative ActA fluorescence intensity at the bacterial surface of V2 and V2 + gshF complemented strain at 6 h p.i. in JEG-3 cells, relative to EGDe-ΔinlA (MOI = 0.5). Each point represents the average ActA intensity per bacterium from a high magnification image of an infection focus. Bars represent the mean ± SEM from 8 isolated foci from two independent experiments. **F**. Low magnification micrographs of dissemination foci at 24 h p.i. in JEG-3 cells infected with EGDe-ΔinlA and V2 (MOI = 0.001). Bacteria appear in white. Bar: 100 µm. **G**. Histogram showing the area of infection foci at 24 h p.i. in JEG-3 cells infected with V2, relative to EGDe-ΔinlA. Each point represents an individual focus taken at low magnification. Bars represent the mean ± SEM of 20 foci per condition from two independent experiments. **H**. High magnification micrographs of JEG-3 cells at 72 h p.i. with EGDe-ΔinlA, EGDe-ΔactA, V2, V2-ΔactA and the complemented strains V2 + gshF and V2-ΔactA + gshF. Insets show high-magnification views of the boxed regions with representative LisCVs. Images show overlays of Hoechst (blue), Lm (green), LAMP1 (red) and F-actin (white) signals. Bars: 10µm. **I**. Histogram of the size of LisCVs at 72 h p.i. in JEG-3 cells infected with EGDe-ΔinlA, EGDe-ΔactA, V2, V2 + gshF, V2-ΔactA and V2-ΔactA + gshF (MOI = 0.01). Each point represents the average size of LisCVs from one microscopic field. The bars represent the mean ± SEM from 4 to 8 microscopic fields per condition from two independent experiments.

ActA expression on bacterial surface, (Fig 3D and 3E). Furthermore, the size of LisCVs, which was increased in V2, returned to a WT size by re-expression of gshF (Fig 3H and 3I). To finally demonstrate that V2 phenotype results of ActA downregulation, we deleted actA in both the V2 strain and the V2 strain expressing gshF. As expected, deletion of actA in the V2 background phenocopied both the ΔactA and V2 phenotypes with respect to LisCV size (Fig 3H and 3I). Importantly, expression of gshF in the V2 strain was no longer able to restore LisCV size in the absence of actA (Fig 3H and 3I).

Taken together, these results identify GshF as a negative regulator of vacuolar persistence, through its positive control of ActA synthesis.

## V3 and V4 are specifically altered at the late persistence phase

Next, we next analyzed the V3 and V4 strains in more details. We first confirmed that both strains did not show any growth defect compared to EGDe-ΔinlA in rich (S6A Fig) or chemically defined media (S6B Fig). Unlike V2, V3 and V4 appeared to form smaller LisCVs compared to EGDe-ΔinlA during long-term infection (Fig 4A). Additionally, the LisCVs generated by V3 and V4 contained fewer bacteria (Fig 4B). Since bacterial capture inside LisCVs requires the disappearance of ActA from the bacterial surface, we investigated whether V3 and V4 might have an altered expression of ActA at 72 h p.i. Immunostaining of ActA at the bacterial surface showed a higher proportion of ActA-positive bacteria in strains V3 and V4, whereas most EGDe-ΔinlA bacteria were ActA negative or trapped inside LisCVs (Fig 4C and 4D). Additionally, V3 and V4 also exhibited significantly higher levels of ActA at their surface (Fig 4E). We next assessed the behavior of V3 and V4 during infection in more detail, by monitoring intracellular CFU over time. Bacterial entry and multiplication were similar between the reference strain and both variants. Surprisingly, at 24 h p.i., the number of intracellular bacteria decreased in V3 and V4 by almost half log compared to the reference strain and remained lower at 72 h (Fig 4F). Microscopic observation revealed that V3 and V4 had a spreading defect compared to EGDe-ΔinlA (Fig 4G). Indeed, V3 and V4 formed a significantly smaller infection foci than the reference strain (Fig 4H). These results are in apparent contrast with a higher amount of ActA at the bacterial surface of V3 and V4. As the polar localization of ActA plays a key role in the ability of Lm to move within host cells, we examined the distribution of ActA on the bacterial surface and found that although the protein is present at higher levels (Fig 4E), its polar repartition is conserved (**Fig 4I**). We conclude that despite having higher levels of ActA on their surface, the activity of ActA in V3 and V4 might be compromised. An overproduction of a dysfunctional ActA is likely to protect bacteria from targeting to LisCVs thus promoting a hypo-vacuolar phenotype (i.e., less associated to LisCVs).

**Table 2. Panaroo analysis between 20 CC9-WT strains and V2 (03EB120LM).**

| VARIANT | PRES/ABS | ANNOTATION | No. ISOLATEs | V2 (03EB120LM) | LOCUSTAG | |
|---|---|---|---|---|---|---|
| group_870 | Present in ALL except 03EB120LM | hypothetical protein | 20 | – | lmo0082 | **Deletion 1** |
| adhR_4 | Present in ALL except 03EB120LM | HTH-type transcriptional regulator AdhR | 20 | – | lmo0083 | **Deletion 1** |
| iolS | Present in ALL except 03EB120LM | Aldo-keto reductase IolS | 20 | – | lmo0084 | **Deletion 1** |
| group_1226 | Present in ALL except 03EB120LM | hypothetical protein | 20 | – | lmo0085 | **Deletion 1** |
| group_1306 | Present in ALL except 03EB120LM | hypothetical protein | 20 | – | lmo0086 | **Deletion 1** |
| group_25 | Present in ALL except 03EB120LM | hypothetical protein | 20 | – | lmo0087 | **Deletion 1** |
| atpE_2 | Present in ALL except 03EB120LM | ATP synthase subunit c | 20 | – | lmo0088 | **Deletion 1** |
| atpD_2 | Present in ALL except 03EB120LM | ATP synthase subunit delta | 20 | – | lmo0089 | **Deletion 1** |
| atpA_2 | Present in ALL except 03EB120LM | ATP synthase subunit alpha | 20 | – | lmo0090 | **Deletion 1** |
| atpG_2 | Present in ALL except 03EB120LM | ATP synthase gamma chain sodium ion specific | 20 | – | lmo0091 | **Deletion 1** |
| atpD_2;atpD_3 | Present in ALL except 03EB120LM | ATP synthase subunit beta;ATP synthase subunit beta sodium ion specific | 20 | – | lmo0092 | **Deletion 1** |
| atpC_2 | Present in ALL except 03EB120LM | ATP synthase epsilon chain | 20 | – | lmo0093 | **Deletion 1** |
| group_522 | Present in ALL except 03EB120LM | hypothetical protein | 20 | – | lmo0094 | **Deletion 1** |
| group_1424 | Present in ALL except 03EB120LM | hypothetical protein | 20 | – | lmo1655 | **Deletion 2** |
| group_689 | Present in ALL except 03EB120LM | hypothetical protein | 20 | – | lmo1656 | **Deletion 2** |
| group_432 | Present in ALL except 03EB120LM | hypothetical protein | 20 | – | lmo2767 | **Deletion 3** |
| group_175 | Present in ALL except 03EB120LM | hypothetical protein | 20 | – | lmo2768 | **Deletion 3** |
| ytrB_3;btuD_5;-ytrB_4 | Present in ALL except 03EB120LM | ABC transporter ATP-binding protein YtrB;Vitamin B12 import ATP-binding protein BtuD | 20 | – | lmo2769 | **Deletion 3** |
| **gshAB** | **Present in ALL except 03EB120LM** | **Bifunctional glutamate--cysteine ligase/ glutathione synthetase** | **20** | **–** | **lmo2770** | **Deletion 3** |
| bglA | Present in ALL except 03EB120LM | 6-phospho-beta-glucosidase BglA | 20 | – | lmo2771 | **Deletion 3** |
| bglF_3 | Present in ALL except 03EB120LM | PTS system beta-glucoside-specific EIIBCA component | 20 | – | lmo2772 | **Deletion 3** |
| licT_2 | Present in ALL except 03EB120LM | Transcription antiterminator LicT | 20 | – | lmo2773 | **Deletion 3** |
| yknY_2;lolD | Present in ALL except 03EB120LM | hypothetical protein | 20 | | lmo2774 | **Deletion 3** |
| group_31 | Present in ALL except 03EB120LM | hypothetical protein | 20 | | lmo2775 | **Deletion 3** |

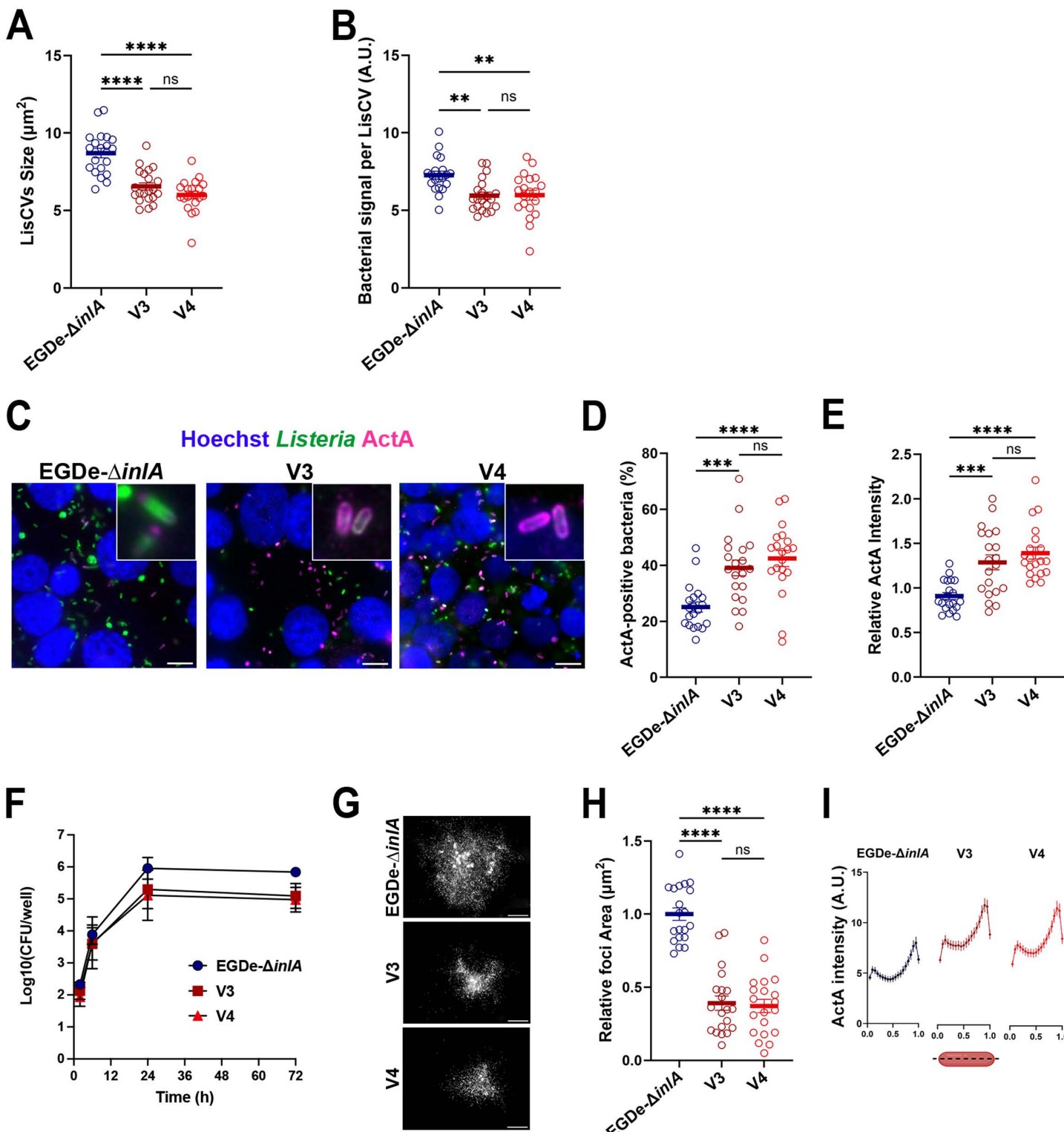

**Fig 4. V3 and V4 share a similar "hypo-vacuolar" phenotype. A**. Histogram showing the size of LisCVs at 72 h p.i. in JEG-3 cells infected with EGDe-Δ*inlA*, V3 and V4 (MOI = 0.001). Each point represents the average size of LisCVs from a microscopic field. Bars represent the mean ± SEM from 20 microscopic fields per condition from two independent experiments. **B**. Average bacterial signal per LisCV (A.U.) at 72 h p.i. in JEG-3 cells

infected with EGDe-Δ*inlA*, V3 and V4. Each point represents the average bacterial signal per LisCV within a single microscopic field. Bars represent the mean ± SEM from 20 microscopic fields per condition from two independent experiments. **C**. Representative micrographs of JEG-3 cells infected with EGDe-Δ*inlA*, V3 and V4 at 72 h p.i. Images show overlays of Hoechst (blue), *Lm* (green) and ActA (magenta). Bars: 10 μm. **D**. Proportion of ActA-positive bacteria at 72 h p.i. in JEG-3 cells infected with EGDe-Δ*inlA*, V3 and V4 (MOI = 0.001). Bars represent the mean ± SEM from 20 microscopic fields from two independent experiments. **E**. Relative ActA fluorescence intensity at the bacterial surface in V3 and V4 at 72 h p.i. in JEG-3 cells, relative to EGDe-Δ*inlA*. Each point represents the average ActA intensity per bacterium in a microscopic field. Bars represent the mean ± SEM from 20 microscopic fields from two independent experiments. **F**. Intracellular infection kinetics of V3 and V4 in JEG-3 cells (MOI = 0.001) compared to EGDe-Δ*inlA* strain (MOI = 0.001). Each point corresponds to the mean CFU count from two wells. Data are representative of two independent experiments. **G**. Low magnification micrographs of infection foci at 24 h p.i. in JEG-3 cells infected with EGDe-Δ*inlA*, V3 and V4 (MOI = 0.001). Bacteria appear in white. Bar: 100 μm. **H**. Quantification of infection focus area at 24 h p.i. in JEG-3 cells for V3 and V4, relative to EGDe-Δ*inlA*. Each point represents an individual focus imaged at low magnification. Bars represent the mean ± SEM from 20 foci per condition from two independent experiments. **I**. Distribution pattern of ActA along the bacterial surface of EGDe-Δ*inlA*, V3 and V4.

## Identification of the genetic determinants responsible for the "hypo-vacuolar" phenotype of V3 and V4

We then sought to identify the genetic variations responsible for the hypo-vacuolar phenotype of V3 and V4. We first examined the sequence of the LIPI-1 locus (S3 Table). No mutations in the virulence genes were found, except for a deletion in *actA* leading to a 35 aminoacid truncated ActA isoform. However, this *actA* allelic variation was present in six other strains of our second screening that showed no effect in both bacterial dissemination and LisCV formation (Fig 2B and S2 Table). Moreover, this mutation was previously described as having no effect on virulence [52–54]. We concluded that additional genetic alterations should be responsible for the hypo-vacuolar phenotype of V3 and V4. We thus employed a comparative genomics approach to find V3- and V4-specific genetic features. We selected EGDe and the 19 InlA-Δ CC9 strains previously screened with V3 and V4 that showed a phenotype similar to the reference strains (EGDe and EGDe-Δ*inlA*) in terms of CFU count at 72 h p.i. and LisCV biogenesis (Fig 2A and 2B and S2 Table). Panaroo analysis did not reveal any missing gene in V3 and V4 compared to the WT phenotype strains. On the other hand, SNIPPY, a software tool used to identify single nucleotide polymorphisms (SNPs) (SNIPPY, https://github.com/tseemann/snippy) [55], identified 22 V3-specific SNPs (19 missense mutations, 1 in-frame deletion, 2 PMSC; Table 3) and 22 V4-specific SNPs (19 missense mutations, 2 frameshifts and 1 PMSC; Table 4). By crossing the V3- and V4-specific SNPs, we found 7 SNPs shared by both strains (Table 5). This indicated that one or more of only 7 genetic DNA sequence polymorphisms may be implicate in the hypo-vacuolar phenotype of V3 and V4.

We next analyzed the functions of these 7 mutated genes. One SNP resulting in a PMSC was located in the *lmo0444* (*yueB*) gene, which is part of the stress island SSI-1. However, SSI-1 is only present in a limited subset of strains (strains 1/2c, 3b and 3c) [56]. Yet, as shown above, LisCV formation is a general phenomenon of the *Lm* species. *lmo0444* therefore did not appear to be a good candidate in the observed phenotype. Five SNPs were missense mutations in genes involved in metabolic pathways: *lmo0078* coding for a phosphoglycerate dehydrogenase; *lmo0224* (*sul*/*folP*), involved in folate biosynthesis; *lmo1845* encoding a guanine hypoxanthine permease; *lmo1915* (*maeA*/*mleS*), encoding a putative NAD-dependent oxaloacetate-decarboxylating malate dehydrogenase; and *lmo2667* (*mtlF*) encoding the EIIA component of the phosphoenolpyruvate:carbohydrate phosphotransferase system (PTS) *Lmo2665–2667* that transports both D-arabitol and D-xylitol [57]. Finally, one SNP was a missense mutation in *lmo2033* (*ftsA*), encoding a major bacterial division factor. The polymorphism in the *lmo0078* gene is unlikely to contribute to the observed phenotype, as one of the strains tested in the initial screen (10CEB615LM, Fig 1A) carries a truncated version of this gene yet displays a WT phenotype (Figs 1B, 1C and S1 Fig).

To understand the contribution of the five remaining sequence polymorphisms in the hypo-vacuolar phenotype of V3 and V4, we replaced each gene on the chromosome with the respective mutated variants in the EGDe-Δ*inlA* strain via allelic exchange. Using this strategy, we could obtain four EGDe-Δ*inlA* recombinant strains. We could not generate the strains harboring the mutated form of *ftsA* possibly because of its essentiality [58]. To circumvent this problem, we constructed an EGDe-Δ*inlA* strain ectopically expressing the mutated form of *ftsA* under the control of a constitutive promoter.

PLOS Pathogens

**Table 3. SNIPPY Analysis between 19 CC9-WT and V3 (17SEL106LM).**

| CHROM | POSI-TION | TYPE | REF (aa) | VAR (aa) | FTYPE | STRAND | NT_POS | AA_POS | EFFECT | LOCUS_TAG | GENE | PRODUCT |
|---|---|---|---|---|---|---|---|---|---|---|---|---|
| 17SEL106LM_4 | 2855563 | snp | G | A | CDS | + | 536/957 | 179/318 | missense_variant c.536G>A p.Arg179Gln | lmo0078 | – | Putative 2-hydroxyacid dehydrogenase |
| 17SEL106LM_4 | 60014 | snp | C | T | CDS | + | 533/786 | 178/261 | missense_variant c.533C>T p.Ser178Leu | lmo0224 | folP | Dihydropteroate synthase |
| 17SEL106LM_4 | 315918 | snp | A | G | CDS | + | 419/1551 | 140/516 | missense_variant c.419A>G p.Asn140Ser | lmo0458 | apc1 | Acetophenone carboxylase alpha subunit |
| 17SEL106LM_4 | 654699 | snp | T | C | CDS | – | 380/471 | 127/156 | missense_variant c.380A>G p.Lys127Arg | lmo0804 | – | hypothetical protein |
| 17SEL106LM_4 | 991276 | snp | A | G | CDS | + | 696/1410 | 232/469 | missense_variant c.696A>G p.Ile232Met | lmo1165 | sucD | Succinate-semialdehyde dehydrogenase (acetylating) |
| 17SEL106LM_4 | 1037296 | snp | G | A | CDS | + | 331/1080 | 111/359 | missense_variant c.331G>A p.Ala111Thr | lmo1217 | ysdC_2 | Putative aminopeptidase YsdC |
| 17SEL106LM_4 | 1058326 | snp | A | G | CDS | + | 2203/2358 | 735/785 | missense_variant c.2203A>G p.Arg735Gly | lmo1232 | mutS2 | Endonuclease MutS2 |
| 17SEL106LM_4 | 1443841 | snp | C | T | CDS | – | 235/822 | 79/273 | missense_variant c.235G>A p.Glu79Lys | lmo1564 | mutM | Formamidopyrimidine-DNA glycosylase |
| 17SEL106LM_4 | 1507491 | snp | T | C | CDS | + | 638/831 | 213/276 | missense_variant c.638T>C p.Met213Thr | lmo1622 | nnrD | ADP-dependent (S)-NAD(P)H-hydrate dehydratase |
| 17SEL106LM_4 | 1752119 | snp | T | A | CDS | – | 1261/1284 | 421/427 | missense_variant c.1261A>T p.Met421Leu | lmo1845 | pbuO_1 | Guanine/hypoxanthine permease PbuO |
| 17SEL106LM_4 | 1820304 | snp | A | G | CDS | + | 1030/1644 | 344/547 | missense_variant c.1030A>G p.Arg344Gly | lmo1915 | mleS | Malolactic enzyme |
| 17SEL106LM_4 | 1918962 | snp | A | C | CDS | – | 944/1305 | 315/434 | missense_variant c.944T>G p.Phe315Cys | lmo2013 | – | hypothetical protein |
| 17SEL106LM_4 | 1945171 | snp | A | G | CDS | – | 956/1281 | 319/426 | missense_variant c.956T>C p.Leu319Pro | lmo2033 | ftsA | Cell division protein FtsA |
| 17SEL106LM_4 | 2051013 | snp | C | T | CDS | – | 5/855 | 2/284 | missense_variant c.5G>A p.Cys2Tyr | lmo2134 | gatY_3 | D-tagatose-1,6-bisphosphate aldolase subunit GatY |
| 17SEL106LM_4 | 2088566 | snp | A | G | CDS | – | 299/1074 | 100/357 | missense_variant c.299T>C p.Phe100Ser | lmo2174 | pleD | Response regulator PleD |
| 17SEL106LM_4 | 2328223 | snp | A | C | CDS | – | 215/858 | 72/285 | missense_variant c.215T>G p.Leu72Arg | lmo2436 | licT_1 | Transcription anti-terminator LicT |
| 17SEL106LM_4 | 2378280 | snp | T | C | CDS | – | 928/1476 | 310/491 | missense_variant c.928A>G p.Thr310Ala | lmo2479 | – | hypothetical protein |

*(Continued)*

**Table 3.** (Continued)

| CHROM | POSI-TION | TYPE | REF (aa) | VAR (aa) | FTYPE | STRAND | NT_POS | AA_POS | EFFECT | LOCUS_TAG | GENE | PRODUCT |
|---|---|---|---|---|---|---|---|---|---|---|---|---|
| 17SEL106LM_4 | 2403590 | snp | G | A | CDS | – | 38/1449 | 13/482 | missense_variant c.38C>T p.Thr13Ile | lmo2503 | clsA_1 | Major cardiolipin synthase ClsA |
| 17SEL106LM_4 | 2558818 | snp | C | T | CDS | – | 119/465 | 40/154 | missense_variant c.119G>A p.Gly40Asp | lmo2667 | mtlF | Mannitol-specific phosphotransferase enzyme IIA component |
| 17SEL106LM_4 | 1853322 | ins | T | TCTG | CDS | – | 853/1788 | 285/595 | conservative_inframe_insertion c.851_853dup-CAG p.Pro284_Glu285insAla | lmo1947 | resE_2 | Sensor histidine kinase ResE |
| 17SEL106LM_4 | 1378787 | snp | A | C | CDS | – | 1141/1143 | 381/380 | stop_lost&splice_region_variant c.1141T>G p.Ter381Gluext*? | lmo1506 | – | hypothetical protein |
| 17SEL106LM_4 | 297312 | snp | T | G | CDS | + | 1888/1890 | 630/629 | stop_lost&splice_region_variant c.1888T>G p.Ter630Gluext*? | lmo0444 | yueB_1 | ESX secretion system protein YueB |

We then tested each recombinant strain by measuring CFU at 72 h p.i. Strikingly, we observed that only the *folP* mutant had a reduced intracellular bacterial load at 72 p.i., similar to V3 and V4 (Fig 5A). Of note, the *folP* mutant multiplied similar to control strain EGDe-Δ*inlA* in rich (S6A Fig) or chemically defined media (S6B Fig).

Altogether, these results reveal that a single aminoacid change from leucine to serine at position 188 in the FolP protein, hereafter referred as FolP$_{L188S}$, phenocopied the replication defect of V3 and V4.

### A defective metabolism of folates impairs *L. monocytogenes* persistence

We then examined whether the EGDe-Δ*inlA-folP*$_{L188S}$ mutant had also defects in LisCVs formation and intracellular motility similar to V3 and V4. Like V3 and V4, EGDe-Δ*inlA-folP*$_{L188S}$ was found in smaller LisCVs (Fig 5B) containing fewer bacteria (Fig 5C) than those generated by EGDe-Δ*inlA*. Moreover, EGDe-Δ*inlA-folP*$_{L188S}$ did not disseminate efficiently and formed smaller foci from 24 h p.i., with foci area being approximately 5-fold smaller than the reference strain (Fig 5D and 5E). Immunostaining and quantification of ActA at the bacterial surface showed that EGDe-Δ*inlA-folP*$_{L188S}$ cytosolic bacteria produced significant more ActA than control EGDe-Δ*inlA* bacteria (Fig 5F and 5G).

Given the increased surface expression of ActA together with the reduced bacterial dissemination and smaller foci formed by the EGDe-Δ*inlA-folP*$_{L188S}$ mutant (Fig 5D-5G), we next assessed whether actin-based motility was functionally impaired. Live cell imaging on JEG-3 cells stably expressing Lifeact-GFP [59] and infected with EGDe-Δ*inlA-folP*$_{L188S}$ showed that, although bacteria were able to recruit actin on their surface, they rarely formed actin tails and were less mobile (S1 and S2 Movies and Fig 5H). Additionally, they were often found in immobile clusters strongly labeled with actin (S1 and S2 Movies and Fig 5I). Together, these data support a model in which the hypo-vacuolar phenotype of V3 and V4 is driven by deregulation of ActA levels and activity downstream of the *folP*$_{L188S}$ mutation. Consistent with this model, *actA* deletion in the *folP*$_{L188S}$ background was epistatic and fully recapitulated the Δ*actA* mutant with respect to LisCV size (Fig 5J).

*folP* has been shown to be essential for *Lm* growth *in vitro* [58]. Our findings suggest that, although the *folP*$_{L188S}$ mutation introduces a strong change in the amino acid sequence of the protein, the catalytic activity of the enzyme should not be completely abolished. To test this hypothesis, the total folate content in EGDe-Δ*inlA*, V3, V4 and EGDe-Δ*inlA-folP*$_{L188S}$

**Table 4. SNIPPY Analysis between 19 CC9-WT and V4 (14SEL860LM).**

| CHROM | POSITION | TYPE | REF (aa) | VAR (aa) | FTYPE | STRAND | NT_POS | AA_POS | EFFECT | LOCUS_TAG | GENE | PRODUCT |
|---|---|---|---|---|---|---|---|---|---|---|---|---|
| 14SEL860LM_1 | 2817613 | snp | G | A | CDS | + | 536/957 | 179/318 | missense_variant c.536G>A p.Arg179Gln | lmo0078 | – | Putative 2-hydroxyacid dehydrogenase |
| 14SEL860LM_1 | 2877885 | snp | T | G | CDS | + | 767/1305 | 256/434 | missense_variant c.767T>G p.Val256Gly | lmo0140 | – | Ribonuclease |
| 14SEL860LM_1 | 61265 | snp | C | T | CDS | + | 533/786 | 178/261 | missense_variant c.533C>T p.Ser178Leu | lmo0224 | folP | Dihydropteroate synthase |
| 14SEL860LM_1 | 76083 | snp | A | C | CDS | + | 79/1374 | 27/457 | missense_variant c.79A>C p.Thr27Pro | lmo0233 | – | hypothetical protein |
| 14SEL860LM_1 | 346690 | snp | T | C | CDS | + | 1405/1995 | 469/664 | missense_variant c.1405T>C p.Ser469Pro | lmo0489 | – | NADH oxidase |
| 14SEL860LM_1 | 394531 | snp | T | C | CDS | – | 607/969 | 203/322 | missense_variant c.607A>G p.Lys203Glu | lmo0535 | ccpB_1 | Catabolite control protein B |
| 14SEL860LM_1 | 538301 | snp | T | G | CDS | + | 285/1224 | 95/407 | missense_variant c.285T>G p.Ile95Met | lmo0681 | flhF | Flagellar biosynthesis protein FlhF |
| 14SEL860LM_1 | 1030619 | snp | A | G | CDS | + | 757/1536 | 253/511 | missense_variant c.757A>G p.Asn253Asp | lmo1208 | cobQ | Cobyric acid synthase |
| 14SEL860LM_1 | 1380155 | snp | A | C | CDS | + | 317/687 | 106/228 | missense_variant c.317A>C p.Asn106Thr | lmo1507 | srrA_1 | Transcriptional regulatory protein SrrA |
| 14SEL860LM_1 | 1657431 | snp | G | A | CDS | – | 1136/1362 | 379/453 | missense_variant c.1136C>T p.Ala379Val | lmo1751 | rlmCD_2 | 23S rRNA (uracil-C(5))-methyltransferase RlmCD |
| 14SEL860LM_1 | 1752139 | snp | T | A | CDS | – | 1261/1284 | 421/427 | missense_variant c.1261A>T p.Met421Leu | lmo1845 | pbuO_1 | Guanine/ hypoxanthine permease PbuO |
| 14SEL860LM_1 | 1820324 | snp | A | G | CDS | + | 1030/1644 | 344/547 | missense_variant c.1030A>G p.Arg344Gly | lmo1915 | maeA | putative NAD-dependent malic enzyme 2 |
| 14SEL860LM_1 | 1945194 | snp | A | G | CDS | – | 956/1281 | 319/426 | missense_variant c.956T>C p.Leu319Pro | lmo2033 | ftsA | Cell division protein FtsA |
| 14SEL860LM_1 | 2296308 | snp | G | A | CDS | – | 2897/3936 | 966/1311 | missense_variant c.2897C>T p.Pro966Leu | lmo2444 | yicI_3 | Alpha-xylosidase |
| 14SEL860LM_1 | 2420313 | snp | A | C | CDS | + | 239/2754 | 80/917 | missense_variant c.239A>C p.Lys80Thr | lmo2558 | inlB_5 | Internalin B precursor |
| 14SEL860LM_1 | 2467255 | snp | T | G | CDS | – | 596/867 | 199/288 | missense_variant c.596A>C p.Lys199Thr | lmo2600 | ecfA2 | Energy-coupling factor transporter ATP-binding protein EcfA2 |
| 14SEL860LM_1 | 2520869 | snp | C | T | CDS | – | 119/465 | 40/154 | missense_variant c.119G>A p.Gly40Asp | lmo2667 | mtlF | Mannitol-specific phosphotransferase enzyme IIA component |

*(Continued)*

**Table 4.** (Continued)

| CHROM | POSI-TION | TYPE | REF (aa) | VAR (aa) | FTYPE | STRAND | NT_POS | AA_POS | EFFECT | LOCUS_TAG | GENE | PRODUCT |
|---|---|---|---|---|---|---|---|---|---|---|---|---|
| 14SEL860LM_1 | 2566231 | snp | T | C | CDS | + | 658/840 | 220/279 | missense_variant c.658T>C p.Tyr220His | lmo2699 | yidA_3 | Sugar phosphatase YidA |
| 14SEL860LM_1 | 2713472 | snp | A | G | CDS | − | 641/795 | 214/264 | missense_variant c.641T>C p.Leu214Pro | lmo2835 | – | D-tagatose 3-epimerase |
| 14SEL860LM_1 | 750574 | ins | A | AG | CDS | + | 197/249 | 66/82 | frameshift_variant c.196dupG p.Glu66fs | lmo0893 | rsbV | Anti-sigma-B factor antagonist |
| 14SEL860LM_1 | 2640995 | ins | C | CA | CDS | − | 344/357 | 115/118 | frameshift_variant c.344dupT p.Leu115fs | lmo2769 | ytrB_4 | ABC transporter ATP-binding protein YtrB |
| 14SEL860LM_1 | 297390 | snp | T | G | CDS | + | 1888/1890 | 630/629 | stop_lost&splice_region_variant c.1888T>G p.Ter630Gluext*? | lmo0444 | – | chromosome segregation protein |

were measured by a microbiological assay (see methods) [60]. We found that V3, V4 and EGDe-Δ$inlA$-$folP_{L188S}$ displayed a similar severely reduced folate content compared to EGDe-Δ$inlA$ (Fig 5K). Complementation of $folP$ restored both LisCVs size and the number of intravacuolar bacteria (Fig 5B and 5C). Moreover, the spreading defect observed in the EGDe-Δ$inlA$-$folP_{L188S}$ as well as the expression of ActA at the bacterial surface were both rescued upon $folP_{WT}$ re-expression (Fig 5D-5G). These phenotypes were likely due to the recovery of intracellular folate levels in the complemented strain (Fig 5K).

Altogether, our results converge toward the conclusion that the intracellular dissemination defect and the hypo-persistence phenotype of V3 and V4 are due to an impaired folate metabolism.

## Discussion

Intracellular pathogens have evolved different strategies to survive inside their host cells. Some can colonize the cytosol while others reside in host-derived vacuoles. Some bacteria can even adopt a dual life style, including both vacuolar and cytosolic stages [12,61]. Although considered for a long time a cytosolic pathogen, several reports have shown that $Lm$ can also inhabit vacuolar compartments [8,9,11]. We have previously shown that during long-term infection in epithelial cells, $Lm$ enters a persistence phase within LAMP1-positive acidic vacuoles that we have named LisCVs [11]. Whether this persistent phenotype was restricted to laboratory strains (EGDe and 10403S) or was a general feature of $Lm$ remained to be determined. In the present study, by using a microscopy-based phenotypic characterization, we assessed the ability of a panel of 105 $Lm$ isolates from 23 different CCs and origins to persist inside LisCVs. We complemented this analysis by quantitative data on bacterial entry and dissemination.

Our results show that the majority of $Lm$ strains (96%) reached the LisCV stage with the same efficiency, regardless their origin (Figs 1C and 2B). Our data are consistent with persistence inside LisCVs as being a common property of $Lm$. Although rare, we could nevertheless identify phenotypic variants with hyper and hypo-vacuolar phenotypes (i.e., more and less associated to LisCVs, respectively) (Figs 1D and 2C). Interestingly, these variants all belong to the hypo-invasive CC9 or CC121, and presented a mutated $inlA$ allele, resulting in the production of a truncated and secreted protein (Figs 1A and 2A and S1 and S2 Tables). This result suggests that bacterial genes playing a role in the intracellular lifestyle, including vacuolar persistence, may have been counter-selected in hypo-invasive strains (i.e., strains that have lost the

**Table 5. Conserved mutations between V3 (17SEL106LM) and V4 (14SEL860LM).**

| CHROM | POSI-TION | TYPE | REF (aa) | VAR (aa) | FTYPE | STRAND | NT_POS | AA_POS | EFFECT | LOCUS_TAG | GENE | PRODUCT |
|---|---|---|---|---|---|---|---|---|---|---|---|---|
| 14SEL860LM_1 | 2817613 | snp | G | A | CDS | + | 536/957 | 179/318 | missense_variant c.536G>A p.Arg179Gln | lmo0078 | – | Putative 2-hydroxyacid dehydrogenase |
| 14SEL860LM_1 | 61265 | snp | C | T | CDS | + | 533/786 | 178/261 | missense_variant c.533C>T p.Ser178Leu | lmo0224 | folP | Dihydropteroate synthase |
| 14SEL860LM_1 | 1752139 | snp | T | A | CDS | – | 1261/1284 | 421/427 | missense_variant c.1261A>T p.Met421Leu | lmo1845 | pbuO_1 | Guanine/ hypoxanthine permease PbuO |
| 14SEL860LM_1 | 1820324 | snp | A | G | CDS | + | 1030/1644 | 344/547 | missense_variant c.1030A>G p.Arg344Gly | lmo1915 | maeA | putative NAD-dependent malic enzyme 2 |
| 14SEL860LM_1 | 1945194 | snp | A | G | CDS | – | 956/1281 | 319/426 | missense_variant c.956T>C p.Leu319Pro | lmo2033 | ftsA | Cell division protein FtsA |
| 14SEL860LM_1 | 2520869 | snp | C | T | CDS | – | 119/465 | 40/154 | missense_variant c.119G>A p.Gly40Asp | lmo2667 | mtlF | Mannitol-specific phosphotransferase enzyme IIA component |
| 14SEL860LM_1 | 297390 | snp | T | G | CDS | + | 1888/1890 | 630/629 | stop_lost&splice_region_variant c.1888T>G p.Ter630Gluext*? | lmo0444 | NA | chromosome segregation protein |

ability to enter cells). However, more strains from CC9 and CC121 will need to be tested in the future to further evaluate this possibility.

Two of the four isolated mutants (V1 and V2) showed defects in the early stages of infection and had mutations in well-known virulence or virulence-regulatory genes. In V1, we identified a premature stop codon leading to the production of a truncated LLO-Δ472 protein (Tables 1 and 6, S2B Fig) that has not been previously described. Some naturally occurring mutations in *hly* that result in loss of LLO activity have been previously identified, in particular, two mutations leading to truncated forms of the protein (LLO N261* and C484*) [34]. V2 has a "hyper-vacuolar" phenotype with a higher proportion of LisCVs-associated bacteria (Fig 3A and 3B), mostly resulting from the disappearance of ActA from the bacterial surface (Fig 3D and 3E). We identified a deletion of *gshF* (Table 6 and S5 Fig), as a major cause of the ActA deficiency. Two other variants (V3 and V4) were specifically affected in the late phases of the infection (from 24 h p.i.) with a defect in LisCV targeting. Indeed, at 72 h p.i., the vacuoles generated by V3 and V4 appeared smaller and contained fewer bacteria than the WT counterpart (Fig 4A and 4B). In addition, V3 and V4 bacteria present in the cytoplasm produced more ActA on their surface (Fig 4C and 4E). Surprisingly, V3 and V4 showed a dissemination defect (Fig 4G and 4H), while the distribution of ActA along the bacterium was not affected (Fig 4I), suggesting an altered function of ActA. Using a comparative genomic analysis, we could link the V3 and V4 hypo-vacuolar phenotype to a mutation in the essential gene *folP* (Table 6 and Fig 5A), which is involved in the synthesis of folic acid and its derivatives. V3 and V4 indeed showed a severely reduced level of intracellular folate (Fig 5J). Interestingly, an impaired folate metabolism seems to primarily affect late stages of infection as V3 and V4 showed a lower bacterial load and defect in intracellular dissemination as early as 24 h p.i. (Fig 4F, 4G and 4H). Folate requirements during *Lm* infection were previously investigated by identifying *Lm* transposon mutants which formed small plaques in tissue culture cells monolayers [62,63]. Two genes encoding enzymes involved in the production of the folic acid precursor, *para*-aminobenzoic acid (PABA) were identified: *pabA* and *pabBC*. Mutants for *pabA* and *pabBC*

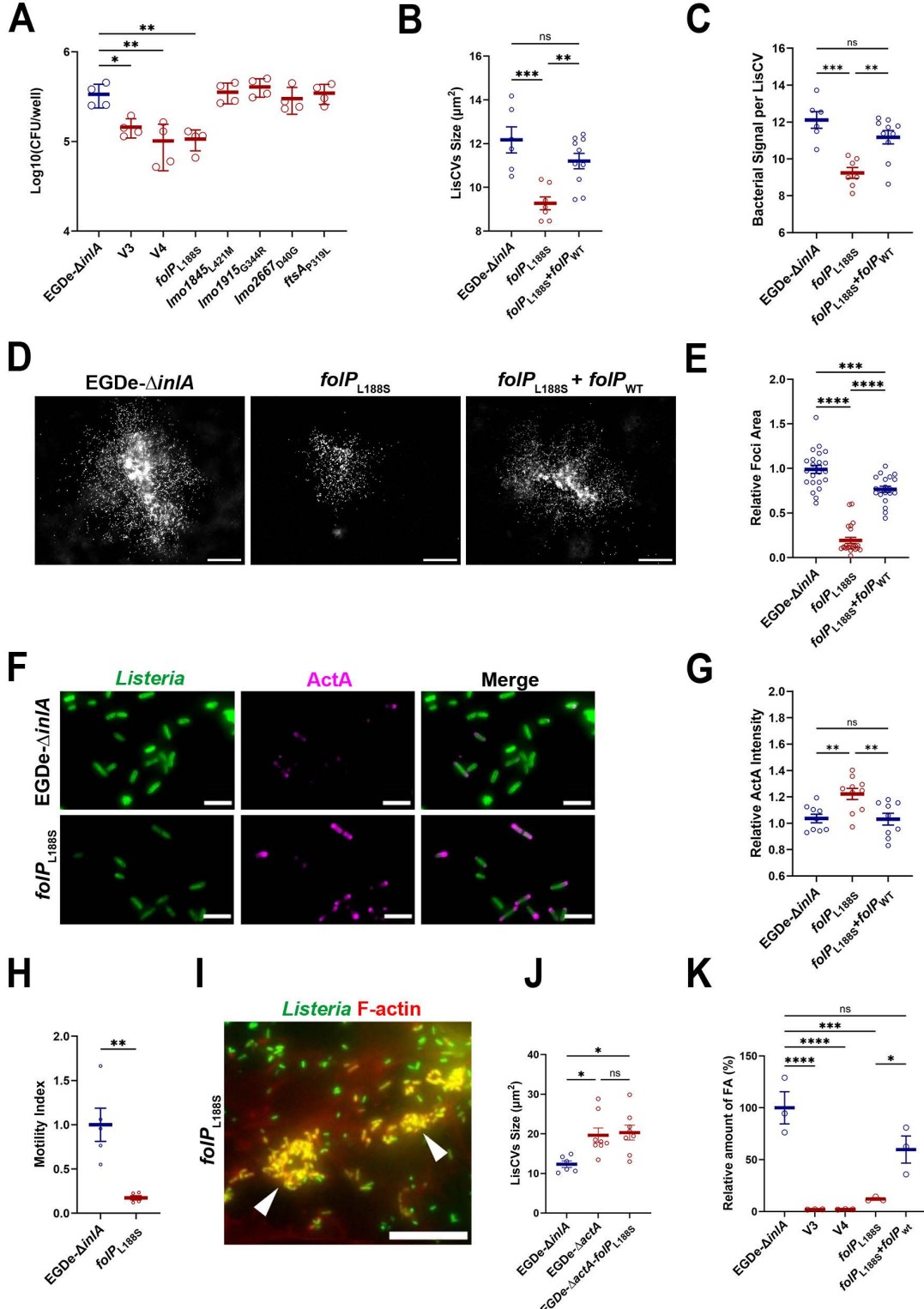

**Fig 5. V3 and V4 have a severely impaired intracellular folate content. A**. Histogram showing the number of intracellular bacteria recovered at 72 h p.i. from JEG-3 cells lysates for EGDe-Δ*inlA*, V3, V4 and the five mutants derived from the EGDe-Δ*inlA* background (MOI = 0.01). Each point represents one infected well. Bars represent the mean ± Standard Deviation (SD) of 4 wells from two independent experiments. **B**. Size of LisCVs

at 72 h p.i. in JEG-3 cells infected with EGDe-Δ*inlA*, EGDe-Δ*inlA-folP*$_{L188S}$ mutant and EGDe-Δ*inlA-folP*$_{L188S}$+*folP*$_{WT}$ complemented strain. Each point represents the average LisCVs size per microscopic field. Bars represent the mean±SEM from 20 microscopic fields per condition across two independent experiments. **C**. Average bacterial signal (A.U.) per LisCV at 72 h p.i. in JEG-3 cells infected with EGDe-Δ*inlA*, EGDe-Δ*inlA-folP*$_{L188S}$ mutant and EGDe-Δ*inlA-folP*$_{L188S}$+*folP*$_{WT}$ complemented strain. Each point represents the average bacterial signal per LisCV per microscopic field. Bars represent the mean±SEM of 20 microscopic fields per condition from two independent experiments. **D**. Representative micrographs of infection foci at 24 h p.i. in JEG-3 cells infected with EGDe-Δ*inlA*, EGDe-Δ*inlA-folP*$_{L188S}$ mutant and EGDe-Δ*inlA-folP*$_{L188S}$+*folP*$_{WT}$ complemented strain (MOI = 0.001). Bacteria are shown in white. Bar: 100 µm. **E**. Quantification of infection focus area at 24 h p.i. in JEG-3 cells infected with EGDe-Δ*inlA-folP*$_{L188S}$ mutant relative to EGDe-Δ*inlA* and EGDe-Δ*inlA-folP*$_{L188S}$+*folP*$_{WT}$ complemented strain. Each point represents one isolated focus imaged at low magnification. Bar represent the mean±SEM of 20 foci per condition from two independent experiments. **F**. Representative micrographs of *Lm* (green) EGDe-Δ*inlA* and EGDe-Δ*inlA-folP*$_{L188S}$ mutant at 72 h p.i. in JEG-3 cells (MOI = 0.01), with ActA protein labeled in magenta. Bars: 5 µm. **G**. Relative ActA fluorescence intensity at the surface of EGDe-Δ*inlA-folP*$_{L188S}$ mutant bacteria at 72 h p.i. in JEG-3 cells, relative to EGDe-Δ*inlA* and EGDe-Δ*inlA-folP*$_{L188S}$+*folP*$_{WT}$ complemented strain. Each point represents the average ActA fluorescence intensity per bacterium within a microscopic field. Bars represent the mean±SEM of 20 microscopic fields per condition from two independent experiments. **H**. Motility index in JEG-3 cells infected with EGDe-Δ*inlA* and EGDe-Δ*inlA-folP*$_{L188S}$ mutant. **I**. High magnification micrographs of JEG-3 cells infected with EGDe-Δ*inlA-folP*$_{L188S}$ mutant at 72 h p.i. Images show overlays of *Lm* (green) and F-actin (red) signals. Arrows indicate immobile bacterial clusters strongly labeled with actin. Bars: 20µm. **J**. Size of LisCVs at 72 h p.i. in JEG-3 cells infected with EGDe-Δ*inlA*, EGDe-Δ*actA* and EGDe-Δ*actA-folP*$_{L188S}$ mutant. Each point represents the average LisCVs size per microscopic field. Bars represent the mean±SEM of 6 to 8 microscopic fields per condition across two independent experiments. **K**. Quantification of folic acid (FA) in cell lysates from EGDe-Δ*inlA*, V3, V4, EGDe-Δ*inlA-folP*$_{L188S}$ mutant and EGDe-Δ*inlA-folP*$_{L188S}$+*folP*$_{WT}$ complemented strain, measured by microbiological assay and normalized to total proteins amount. Each point represents one independent stationary-phase BHI culture for each strain. Bar represent the mean±SD of 3 cultures from three independent experiments.

**Table 6. Genetic characteristics of strains affecting LisCV formation.**

| Strain | Mutation type | Gene/ position | Predicted effect | Vacuole phenotype |
|---|---|---|---|---|
| 2965 (V1) | Single nucleotide insertion | hly (c.1383_1384insA), LLO-Δ472 | Frameshift, PMSC→Loss of function | Complete loss |
| 03EB120LM (V2) | Deletion | gshF | Loss of function | Large vacuoles |
| 17SEL106LM (V3) | Missense SNP | sul/folP (c.533T>C), FolP-L188S | Enzymatic alteration→affects folate metabolism | Small vacuoles |
| 14SEL860LM (V4) | Missense SNP | sul/folP (c.533T>C), FolP-L188S | Enzymatic alteration→affects folate metabolism | Small vacuoles |

were severely attenuated for intracellular growth in bone marrow-derived macrophages (BMMs), they produced extremely small plaques in fibroblast monolayers, and were attenuated for virulence in mice [64]. The growth defect of the PABA-starved Δ*pabA* mutant in BMMs was attributed to its inability to propagate in the host cytosol, suggesting that although these bacteria were present in the cytosol, they failed to have sufficient functional ActA to induce host actin polymerization [64]. Production and localization of ActA at the bacterial surface of Δ*pabA* was not investigated. Here, we show that folate deficiency leads to an upregulation of ActA protein level at the bacterial surface (Fig 5F and 5G). However, bacteria show a defect in cell-to-cell spread (Fig 5D and 5E), indicating that the activity of ActA was compromised. Mechanistically, how folate metabolism impact ActA proteins levels and activity remains to be determined. During active phases of dissemination, ActA is concentrated at one pole of the bacterium. This asymmetrical distribution of ActA induces actin filament nucleation and polymerization, which are essential for intracellular motility [65]. During bacterial multiplication, ActA is absent from the septation site and translocates to the "old" pole [66]. We observed that although ActA was more abundant on the bacterial surface of the *folP* mutant strains, the polar localization of ActA appeared to be maintained. Reduced bacterial motility might therefore be due to either a subtle mislocalization of ActA or impaired nucleation activity. Our data revealed that ActA is a major player for vacuolar persistence and its expression at the bacterial surface appears to be sufficient in inhibiting bacterial sequestration inside LisCVs. In agreement with this observation, when the expression level of ActA is compromised bacteria are more efficiently captured inside LisCVs ([11] and Fig 3). Furthermore, ActA-expressing bacteria are less associated to LAMP1, regardless of their motility (S1 and S2 Movies and Fig 4). It is likely that other bacterial genes that

affect either directly or indirectly ActA stability or presentation at the bacterial surface have some impact on the persistence capacity of *Lm.*

Our work reveals that vacuolar persistence is a general property of *L. monocytogenes* and highlights genes supporting the intracellular persistent lifestyle. Genome-wide identification of bacterial determinants involved in LisCVs biogenesis or bacterial survival inside LisCVs will be instrumental to shed light on this poorly characterized intracellular phase that could play an important role in the asymptomatic carriage of *Lm*. The mammalian factors playing a role in their formation also remain to be determined. Deciphering how this intracellular niche is formed and maintained becomes critical to understand the long-term crosstalk between *Lm* and its host.

## Materials and methods

### Human cell lines and bacterial strains

JEG-3 trophoblastic cells (ATCC HBT-3) were maintained at 37°C in a humidified atmosphere containing 5% CO2, in Minimum Essential Medium (MEM) supplemented with GlutaMAX (GIBCO), 2mM glutamine, 1% non-essential amino acids (NEAA), 1mM sodium pyruvate, and 10% fetal calf serum (Sigma). A JEG-3 cell line stably expressing LifeAct-GFP was established following transduction with the lentiviral vector pLenti.PGK.LifeAct-GFP.W (Addgene plasmid #51010), which expresses LifeAct-GFP under the control of the human PGK promoter. *Lm* and *Escherichia coli* strains were cultured at 37°C with agitation in brain-heart infusion (BHI) or Luria-Bertani (LB) broth (BD Difco), respectively. When applicable, antibiotics were added to the media for plasmid maintenance.

A total of 105 *Lm* isolates from the two main lineages (I and II) were analyzed in this study, as detailed in S1 Table (initial screening) and S2 Table (second screening). The *Lm* reference strains, mutant derivatives, and plasmids used are listed in S4 Table.

### Mutant construction

The V2-Δ*actA* was generated by allelic exchange, as described previously [11]. Complemented strains, V2 + *gshF* and V2-Δ*actA* + *gshF*, were generated by electroporating the chromosome-integrative plasmid pPL2-*gshF* into the V2 variant and V2-Δ*actA* mutant, respectively. To create plasmids carrying specific mutated alleles, four contructs, pMAD-*folP*[563T > C], pMAD-*lmo1845*[1261T > A], pMAD-*lmo1915*[1030G > A] and pMAD-*lmo2667*[119A > G] were designed. For each construct, ~ 600 bp fragments upstream and downstream of the targeted mutation site were synthetized and cloned into pMAD vector [67], enabling allelic exchange in the EGDe-Δ*inlA* background. This approach generated the following mutant strains: EGDe-Δ*inlA-folP*$_{L188S}$, EGDe-Δ*inlA-lmo1845*$_{L421M}$, EGDe-Δ*inlA*-lmo*1915*$_{G344R}$, and EGDe-Δ*inlA-lmo2667*$_{D40G}$.

The pAD-*ftsA*[956C > T] plasmid was constructed using pAD-cGFP as a template [68]. A 1300 bp fragment containing the full-length *ftsA* gene carrying the 956C > T substitution was synthetized and inserted into the BglII/SalI sites, enabling ectopic expression of the mutated *ftsA*$_{P319L}$ gene under the control of the P*hyper* promoter. The plasmid was introduced into *Lm* EGDe-Δ*inlA* strain by electroporation.

The pAD-*folP*$_{WT}$ plasmid was constructed using pAD-cGFP as a template [68]. A 922 bp fragment containing the full-length *folP* gene from EGDe was synthetized and cloned into the SacI/SalI sites, allowing ectopic expression under the control of the P*hyper* promoter. The plasmid was introduced into EGDe-Δ*inlA-folP*$_{L188S}$ strain by electroporation, leading to the generation of the EGDe-Δ*inlA-folP*$_{L188S}$+*folP*$_{WT}$ complemented strain. For all mutants, colony PCR was performed to amplify the insertion site, and sequencing confirmed the presence of the desired mutation in the bacterial genome. The DNA fragments were synthetized and cloned into the corresponding vectors by Genecust.

### Testing of hemolytic activity of the V1 variant

The hemolytic activity of the V1 variant was evaluated using Columbia horse blood agar plates (BioMérieux, France), as previously described [34]. For comparison, several control strains were included: *Lm* 17SEL410LM (CC121-WT control

strain) and 10403S, were used as positive controls, while the hemolysin-deficient strain 10403S-Δ*hly* served as negative control.

## Growth curves of bacterial strains

For growth curve analysis, a single colony of each strain was inoculated into 5 mL of BHI broth and incubated at 37°C with agitation. A 1:1000 dilution of the overnight culture was prepared in BHI broth, and 200 μL was dispensed into each well of a 96-well microtiter plate. Bacterial growth was monitored over 24 hours at 600nm and 37°C using a Spark Tecan microplate spectrophotometer.

## Bacterial infections

The long-term infection protocol in JEG-3 cells was performed as previously described in Kortebi et al., 2017 [11] with modifications. JEG-3 cells were seeded in 24-well plates, with or without coverslips, to reach 90–100% confluency on the day of infection. On the day of infection, bacterial inocula were prepared from stationary-phase bacteria grown in BHI overnight, then diluted in serum-free MEM medium.

**Screen 1.** JEG-3 cells were washed with serum-free medium and then infected for 1 h with 1 mL of bacterial suspension at a multiplicity of infection (MOI) of ~ 0.01 bacteria/cell. The plates were centrifuged at 300 x g for 2 min to synchronize the entry of bacteria. After 1h of infection, complete medium containing gentamicin (SIGMA) at 25μg/mL was added to cells to kill extracellular bacteria. Entry assays were performed at a comparable MOI of about 1 bacterium per 100 cells (MOI ~ 0.01). After 2 hours of infection, cells were lysed in cold water and the intracellular bacteria counted on BHI agar plates. The percentage of entry of each strain was determined by relating the number of intracellular bacteria to the number of bacteria in the inoculum from 3 independent experiments and compared to the laboratory strain 10403S, used as a reference (% entry = 100). The total bacterial load per well at 72 h p.i. was determined by CFU/well (n = 3 independent experiments).

**Screen 2.** The MOI was adjusted to reach a similar number of intracellular bacteria (about 20–100) at 2 h p.i., and monitor bacterial dissemination from isolated infected cells to individual infection foci at 72 h p.i. We adjusted the MOI of hypo-invasive CC9 and CC121 (*inlA*-Δ strains) and mutant EGDe-Δ*inlA* to 0.001-0.05 and of control strains EGDe or 10403S (*inlA*-WT) strains to 0.0001–0.0005, to obtain a normalized entry. After 2 h and 72 h of infection, the cells were either lysed in cold water for bacterial enumeration on BHI agar or fixed for immunofluorescence experiments. The total bacterial load per well at 72 h p.i. was determined by CFU/well and are from 3 independent experiments.

## Kinetics of infection

JEG-3 monolayer was infected with bacteria at MOI of 0.001 or 0.01, followed by centrifugation at 300g for 2 min to synchronize bacterial entry. After 1 hour of infection, the medium was replaced with complete medium containing 25 μg/mL gentamicin to eliminate extracellular bacteria. Cells were either fixed for immunofluorescence or lysed at various time points post-infection (2 h, 6 h, 24 h and 72 h) using cold distilled water. The number of viable intracellular bacteria was determined by plating serial dilutions in PBS and enumerating colony-forming units (CFU).

## Antibodies reagents and epifluorescence microscopy

The primary antibodies and fluorescent secondary antibodies used in this study are detailed in Kortebi et al., 2017 [11]. Infected cells were processed for immunofluorescence analysis following protocols previously established in Kortebi et al. (2017) [11]. Briefly, cells were permeabilized with 0.4% Triton X-100, washed, and subjected to immunofluorescence using antibodies diluted in 2% BSA. Fluorescent phalloidin and DAPI (or Hoechst) were used to stain F-actin and nuclei. Samples were mounted on glass slides and analyzed using Carl Zeiss Axiovert 135. Images were acquired at high

magnifications (63x) to visualize the infection process and at low magnification (20x) to visualize the dissemination of the strains. All quantifications have been achieved with ImageJ/Fiji software [69].

### Quantification of LisCVs size and number of bacteria per LisCVs

Stacked images (5 µm depth) were acquired at high magnification (63x), capturing 20–30 nuclei per field. Z-projection were performed for each image. After background subtraction, fluorescence channels 488 and Cy3 were binarized using the *Auto-Threshold* tool to identify bacteria and LAMP1-positive vacuoles, respectively. LisCVs were isolated automatically to keep round-shaped bacterial signal colocalizing with LAMP1 and then sorted manually to eliminate false positive. These LisCVs were saved as Region of Interest (ROI), and their areas were measured. The number of bacteria per LisCVs was estimated based on the intensity of the bacterial green signal within each ROI. For both LisCVs size and bacterial signal per LisCVs, average values were calculated on a whole image.

### Quantification of actin-associated bacteria

Stacked images (5 µm depth) were acquired at high magnification (63x), capturing 20–30 nuclei per field. Z-projections were generated for each image. Actin-associated bacteria were identified manually by selecting bacterial signals colocalizing with polymerized actin structures exhibiting cloud- or comet-like morphologies. The proportion of actin-positive bacteria was calculated relative to total cytosolic bacteria, i.e., those not associated with the LAMP1 signal. Average values were calculated across the entire image and for four images from at least two independent experiments.

### Quantification of foci area

RFP expressing strains were used to infect JEG-3 cells. Images were acquired at low magnification (20x) to observe individual infection foci. After thresholding and binarization, the area of each focus was measured as the sum of the surface area covered by the RFP signal.

### Quantification of the ActA intensity

Stacked images (5 µm depth) were acquired at high magnification (63x), capturing 20–30 nuclei per field. Z-projection were performed for each image. As the target of the anti-*Listeria* antibodies can be masked by the presence of ActA, and therefore actin, on the bacterial surface, both ActA-Cy3 and Listeria-488 channels were merged to reconstitute the full rod-shaped outline of the bacteria. Merge channels images were processed by background subtraction and subsequently binarized. The *Analyze Particles* tool was used to select isolated rod-shaped bacteria, which were then saved as ROI. Within each ROI, the intensity of the ActA-Cy3 signal was quantified. Bacteria were classified as ActA-positive when the ActA signal covered more than 5% of the total area of the bacterial rod. The average ActA intensity per bacterium was calculated based on all ActA-positive bacteria on a same field.

### Construction of ActA distribution profiles

Bacterial cells were segmented using the *Analyze Particle*s tool in ImageJ, and only ActA-expressing bacteria were retained for analysis. The Feret diameter was obtained and used as a proxy for bacterial length, defined as the distance between the two poles of each bacillus. ActA fluorescence intensity was quantified along this axis. To ensure consistent orientation, bacteria were aligned in RStudio such that the pole exhibiting the maximum ActA intensity was positioned uniformly. The profiles were normalized to the length of the bacteria, and an average ActA distribution profile was obtained for each strain by calculating the mean of the ActA intensity at the different relative positions.

## Folic acid microbiological assay

Samples have been collected from 25-mL overnight cultures in BHI by centrifugation. Pellets were resuspended with 1 mL of sterile water. Bacteria were lysed using FastPrep instrument. Lysates were then collected and filter sterilized. Bradford total protein assay was performed as control for the lysis. To assay the amount of folic acid, a microbial assay has been performed using Difco Folic Acid Assay Medium (FAAM) and the strain *Enterococcus hirae* LMG6399 (ATCC 8043) as previously described [60]. Rapidly, *E. hirae* is cultured in FAAM in presence of different dilutions of the lysates (1:50 or 1:200). The cultures are incubated for 24 h at 37°C before measuring the OD at 600 nm. The amount of folic acid in each sample was expressed relative to that in EGDe-Δ*inlA* samples, after normalisation for dilution and total protein concentration in the lysates.

## Statistical analysis

Graphs and statistical analyses were performed using RStudio [70], with the *ggplot2*, *ggprism* and *rstatix* packages in addition to base R functions. Statistical differences between group means were assessed using non-parametric Wilcoxon-Mann-Whitney tests or Kruskal-Wallis tests (non-parametric ANOVA), followed by Dunn's post-hoc pairwise comparisons. For comparisons involving three or more groups, *p*-values were adjusted using Holm's correction method. Statistical significance was defined as an adjusted *p*-values $< 0.05$. Significance levels are indicated as follows: ns: $p > 0.05$; *: $p < 0.05$; **: $p < 0.01$; ***: $p < 0.001$; ****: $p < 0.0001$.

## Genome Sequencing

The raw reads produced by paired-end illumina sequencing were processed using the *in-house* workflow ARTWork (github.com/afelten-Anses/ARtWORK). This workflow performs multiple quality control and curation steps, including read contamination detection, quality assessment, de novo assembly, and annotation, as described in previous studies [71,72]. Briefly, *de novo* assembly was performed with SPAdes v3.9 assembler and contigs were annotated with Prokka v1.13.4. Sequence type (ST) and clonal complexes (CC) were determined based on the Listeria Multilocus sequence typing (MLST) scheme from Institut Pasteur [43]. Accession numbers, typing data, the sequencing coverage and genome assembly metrics are provided in **S2 Table**. All reads passed ARTWork's quality control step and no contamination was detected by Confindr.

## Genetic Basis of Hypo-virulence in V1–V4

To investigate the hypo-virulent phenotype observed in the four strains (V1, V2, V3, and V4) from CC9 and CC121, we applied a stepwise approach. We first screened for known virulence genes, assessing whether truncations or specific mutations could explain the phenotype. This was the case for V1, where a clear genetic alteration was identified, leading us to conclude our analysis at this stage. For the remaining three variants, no obvious virulence gene mutations were found, prompting us to conduct a pangenome analysis to examine gene presence/absence patterns that might account for the phenotype. This approach successfully explained the hypo-virulence of V2. However, for V3 and V4, neither virulence gene screening nor pangenome analysis provided a clear explanation. Consequently, we performed SNP calling across the core genome to identify potential mutations associated with their hypo-virulent phenotype

## Genetic Determinants of Hypo-virulence in V1-V4

Virulence gene analysis was performed using the *Listeria monocytogenes* database within BIGSdb-Pasteur [43], which provides a curated set of alleles for 92 known virulence-associated genes. Sequencing reads or assembled genomes were analyzed by mapping or BLAST-based searches against the virulence gene collection available in BIGSdb. This approach enabled the identification of allelic variants and potential mutations (e.g., premature stop codons or deletions)

that may attenuate virulence, as well as the comparison of virulence gene presence/absence between wild-type and variant strains to assess potential correlations with the observed phenotypes.

To better capture the genetic diversity and to obtain a larger pangenome, Panaroo v 1.2.7 was used with sensitive mode [48] to search the presence-absence of genes associated with the hypo-virulence of our three variants within the CC9 genomes. Differences observed in the gene content between hypo-virulent and wildtypes strains were manually confirmed with a BLAST and mapping analysis.

Variant calling was perform using the bacterial SNP-calling pipeline Snippy (v4.6.0) with default parameters (https://github.com/tseemann/snippy) [55]. To identify mutations (SNPs leading to missense mutations, small deletions-insertions or frameshifts) specific to the CC9 variants, the reads of the 20 wild-type isolates were aligned against the assemblies of 17SEL106LM (V3) and 14SEL860LM (V4).

## Supporting information

**S1 Fig. Identification of a bacterial strain with reduced intracellular multiplication in JEG-3 placental cells.** Entry capacity at 2h p.i. (A) and intracellular multiplication capacity at 72h p.i. (B) were analyzed according to strains, clonal complexes and lineages. Laboratory strains 10403S, EGDe and LO28 are shown in black. The 2965 variant (V1), which exhibits a significant reduction in intracellular bacterial load, is indicated with an asterisk. Results represent the mean of three independent experiments. Statistical significance was determined using a two-Sample t-test comparing each strain to the reference strain10403S, with Holm's adjustment for p-value (*: $p \leq 0.05$).
(TIF)

**S2 Fig. V1 variant exhibits impaired vacuolar escape due to a mutation in LLO.** A. Growth curve of the 2965 variant (V1) in BHI medium, compared to the reference strains EGDe and CC121-WT (17SEL410LM). B. Schematic representation of listeriolysin O (LLO) from the reference strain EGDe (top) and the V1 variant (bottom). The signal peptide, pH sensor (acidic triad D208, E247, D320), and cholesterol-binding motif are indicated, with the two transmembrane β-hairpins (TMH) highlighted in blue. In V1, a mutation at position 462 (red arrow) induces a frameshift (fs), resulting to the loss of the 67 C-terminal amino acids, which are essential for membrane binding and proper LLO function.
(TIF)

**S3 Fig. V2 is strongly impaired in cell-to-cell spread.** High magnification micrographs of JEG-3 cells infected for 6 h with EGDe-ΔinlA (top panel) and V2 (03EB120LM, bottom panel). Images show overlays of Hoechst (blue), Lm (green) and F-actin (red) signals. Bars: 20 μm.
(TIF)

**S4 Fig. InlA-Δ CC9 strains exhibiting a wild-type behavior, similar to the EGDe-ΔinlA reference strain, in actin polymerization (6 h p.i.) and LisCVs formation (72 h p.i.).** A. Representative micrographs of JEG-3 cells infected for 6 h with the indicated strains. Images show overlays of Hoechst (blue), Lm (green) and F -actin (red) signals. Bars: 10μm. Insets highlight high-magnification views of the boxed regions with bacteria forming actin comets. B. Micrographs of JEG-3 cells infected for 72 h with the indicated strain. Images show overlays of Hoechst (blue), Lm (green), LAMP1 (red) and F-actin (white) signals. Bars: 10μm. Insets show high-magnification views of the boxed regions with representative LisCVs.
(TIF)

**S5 Fig. Gene content of the gshF region (lmo2766 to lmo2776) in EGDe compared to the corresponding region in V2 (03EB120LM).** The 9 genes present in EGDe but absent in the V2 variant are shown in green, with the gshF gene, encoding for the glutathione synthase, highlighted in bold. The IS1542, that substitutes the gshF region, is depicted in

orange. The two inverted repeats flanking the transposase are represented by purple rectangles. Genome comparisons were visualized using Easyfig, version 2.1 [73].
(TIF)

**S6 Fig. V3, V4 and EGDe-ΔinlA-folP$_{L188S}$ mutant multiplied similar to control strain EGDe-ΔinlA in BHI (A) or chemically definied medium (B).**
(TIF)

**S1 Table. Characteristics of isolates used in the first screen.**
(XLSX)

**S2 Table. Characteristics of the InlA-Δ isolates used in the second screen.** CC9-WT strains used for genomic analyses are shown in bold.
(XLSX)

**S3 Table. Comparison of the LIPI-1 virulence locus among the EGDe strain, V2, V3 and V4 (03EB120LM, 17SEL106LM and 14SEL860LM).**
(XLSX)

**S4 Table. Bacterial strains and plasmids used in this study.**
(XLSX)

**S1 Movie. Time-lapse images of JEG-3 cells infected with RFP-expressing EGDe-ΔinlA strain.** Infections were performed as described in the Materials and Methods section. Fluorescent signals were acquired from seven Z-planes spanning 7 µm in depth using an autofocus system. One image was captured every 25 seconds, and the videos are shown at approximately 300x real-time speed. Scale bar: 20 µm.
(AVI)

**S2 Movie. Time-lapse images of JEG-3 cells infected with RFP-expressing EGDe-ΔinlA-folP$_{L188S}$ mutant strain.** Infections were performed as described in the Materials and Methods section. Fluorescent signals were acquired from seven Z-planes spanning 7 µm in depth using an autofocus system. One image was captured every 25 seconds, and the videos are shown at approximately 300x real-time speed. Scale bar: 20 µm.
(AVI)

## Acknowledgments
We thank Dan Portnoy for the gift of pPL2-gshF and pPL2-PactA-RFP (G. Mitchell) and Pascale Serror for the gift of *Enterococcus hirae* LMG6399 (ATC 8043). We thank the members of the EpiMic team for helpful discussions.

## Author contributions
**Conceptualization:** Hélène Bierne, Alessandro Pagliuso, Eliane Milohanic.

**Data curation:** Aurélie Lotoux, Matthieu Bertrand, Pierre-Emmanuel Douarre, Mounia Kortebi, Hélène Riveiro, Federica Palma, Goran Lakisic, Edward M. Fox, Laurent Guillier, Hélène Bierne, Alessandro Pagliuso, Eliane Milohanic.

**Formal analysis:** Aurélie Lotoux, Matthieu Bertrand, Pierre-Emmanuel Douarre, Edward M. Fox, Laurent Guillier, Hélène Bierne, Alessandro Pagliuso, Eliane Milohanic.

**Funding acquisition:** Hélène Bierne, Alessandro Pagliuso, Eliane Milohanic.

**Investigation:** Aurélie Lotoux, Matthieu Bertrand, Pierre-Emmanuel Douarre, Mounia Kortebi, Hélène Riveiro, Federica Palma, Goran Lakisic, Edward M. Fox, Laurent Guillier, Hélène Bierne, Alessandro Pagliuso, Eliane Milohanic.

**Methodology:** Aurélie Lotoux, Matthieu Bertrand, Pierre-Emmanuel Douarre, Mounia Kortebi, Hélène Riveiro, Goran Lakisic, Hélène Bierne, Alessandro Pagliuso, Eliane Milohanic.

**Project administration:** Hélène Bierne, Alessandro Pagliuso, Eliane Milohanic.

**Resources:** Edward M. Fox, Laurent Guillier, Anna Oevermann, Sophie Roussel.

**Supervision:** Hélène Bierne, Alessandro Pagliuso, Eliane Milohanic.

**Validation:** Aurélie Lotoux, Matthieu Bertrand, Pierre-Emmanuel Douarre, Mounia Kortebi, Federica Palma, Goran Lakisic, Edward M. Fox, Laurent Guillier, Hélène Bierne, Alessandro Pagliuso, Eliane Milohanic.

**Visualization:** Aurélie Lotoux, Matthieu Bertrand, Pierre-Emmanuel Douarre, Mounia Kortebi, Hélène Bierne, Eliane Milohanic.

**Writing – original draft:** Hélène Bierne, Alessandro Pagliuso, Eliane Milohanic.

**Writing – review & editing:** Aurélie Lotoux, Matthieu Bertrand, Pierre-Emmanuel Douarre, Hélène Bierne, Alessandro Pagliuso, Eliane Milohanic.

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
