## [Decision Letter · Decision Letter 0]

24 Aug 2025

Large-scale phenotyping and comparative genomics reveal genetic features of Listeria persistence in epithelial cells

PLOS Pathogens

Dear Dr. Milohanic,

Thank you for submitting your manuscript to PLOS Pathogens. After careful consideration, we feel that it has merit but does not fully meet PLOS Pathogens's publication criteria as it currently stands. Therefore, we invite you to submit a revised version of the manuscript that addresses the points raised during the review process.

Please submit your revised manuscript within 60 days Oct 23 2025 11:59PM. If you will need more time than this to complete your revisions, please reply to this message or contact the journal office at plospathogens@plos.org. Please include the following items when submitting your revised manuscript:

We look forward to receiving your revised manuscript.

Kind regards,

Alice Prince

Section Editor

PLOS Pathogens

Editor-in-Chief

PLOS Pathogens

orcid.org/0000-0003-2946-9497

Editor-in-Chief

PLOS Pathogens

orcid.org/0000-0002-7699-2064

**Additional Editor Comments:**

As noted by the reviewers - this is comprehensive analysis of the intracellular lifestyle of an important pathogen. The studies are well done and certainly add to our understanding of the intracellular lifestyle of Listeria monocytogenes. However, several suggestions have been made to improve the manuscript which need to be addressed. it is important to document your findings in an in vivo model, as they suggest as well as in CaCO cells and address the numerous points to solidify and clarify the experimental conclusions.

**Journal Requirements:**

At this stage, the following Authors/Authors require contributions: Aurélie Lotoux, Matthieu Bertrand, Pierre-Emmanuel Douarre, Mounia Kortebi, Hélène Riveiro, Federica Palma, Goran Lakisic, Edward M. Fox, Laurent Guillier, Anna Oevermann, Sophie Roussel, Hélène Bierne, Alessandro Pagliuso, and Eliane Milohanic. Please ensure that the full contributions of each author are acknowledged in the "Add/Edit/Remove Authors" section of our submission form.

2) We noticed that you used the phrase 'unpublished ' in the manuscript. We do not allow these references, as the PLOS data access policy requires that all data be either published with the manuscript or made available in a publicly accessible database. Please amend the supplementary material to include the referenced data or remove the references.

- TM on pages: 26, and 29.

Potential Copyright Issues:

i) Figures 1A, and 2A. Please confirm whether you drew the images / clip-art within the figure panels by hand. If you did not draw the images, please provide (a) a link to the source of the images or icons and their license / terms of use; or (b) written permission from the copyright holder to publish the images or icons under our CC BY 4.0 license. Alternatively, you may replace the images with open source alternatives. See these open source resources you may use to replace images / clip-art:

6) Thank you for stating "Genome sequences are deposited and freely available on ncbi (please see supplementary informations). Please specify where the accession numbers can be found in the supplementary information file. Or, you can update your Data Availability Statement to include the DOI/accession number of each dataset.

7) Please amend your detailed Financial Disclosure statement. This is published with the article. It must therefore be completed in full sentences and contain the exact wording you wish to be published.

2) If any authors received a salary from any of your funders, please state which authors and which funders.

8) Please provide a completed 'Competing Interests' statement, including any COIs declared by your co-authors. If you have no competing interests to declare, please state "The authors have declared that no competing interests exist". Otherwise please declare all competing interests beginning with the statement "I have read the journal's policy and the authors of this manuscript have the following competing interests:"

**Reviewers' Comments:**

Reviewer's Responses to Questions

**Part I - Summary**

Reviewer #1: In this study, the authors investigate the ability of Listeria monocytogenes (Lm) strains from diverse clonal complexes (CCs) to form LisCVs, LAMP1-positive vacuoles associated with Lm persistence in epithelial cells. Using 105 isolates of different orgins, they demonstrate that LisCV formation is broadly conserved across CCs, regardless of their source. Among these, three isolates (V2, V3, and V4—all CC9 with inlA PMSC mutations) exhibited altered LisCV phenotypes. The authors show that ectopic expression of gshF in V2 restores ActA expression and increases LisCV size. They also identify a folP L188S mutation in V3 and V4 that reduces Lm proliferation, cell-to-cell spread, and LisCV size, while paradoxically increasing ActA surface expression in EGDe. These findings suggest that ActA regulation plays a key role in Lm persistence, especially among hypo-virulent CC9 strains, while the role of actA in LisCV was already demonstrated previously (Kortebi et al., 2017). As highlighted in the abstract and author summary, the study's central novel finding is the identification of folP as a regulator of ActA expression and function, likely mediated through impaired folate biosynthesis, given that EGDe expressing folP L188S fails to synthesize folate at wild-type levels. However, the underlying mechanism remains unclear.

Reviewer #2: Listeria is now appreciated to persist within host tissue, in part, by growing within vacuolar compartments in host cells. In this manuscript, the authors use a phenotyping and compartative genomics approach to identify genetic features that impact Listeria persistence in epithelial cells. They identify new variants in hly, gshF and folP genes that impact Listeria persistence. Mechanistically they find that ActA expression and function are critical for generation of the persistence niche. The manuscript addresses an important and understudied aspect of Listeria infections and, through the use of many clinical isolates, supports the emerging view that persistence in host cells is a critical virulence trait. The paper is well written and the provides a roadmap for future studies of its kind. I include my comments for the authors consideration.

**Part II – Major Issues: Key Experiments Required for Acceptance**

Please use this section to detail the key new experiments or modifications of existing experiments that should be absolutely required to validate study conclusions.required to validate study conclusions.

Reviewer #1: While elucidating how folP L188S affects ActA function may be challenging, the authors could strengthen their claim by addressing whether ActA is required for LisCV modulation in the isolates (V2, V3 and V4). This could be tested using EGDe folP L188S combined with inlA and actA deletions. A similar approach should be applied to gshF complementation in V2: introducing actA deletion in V2 or in V2 expressing gshF would clarify whether ActA mediates the observed LisCV phenotypes.

All in vitro experiments were performed using JEG3 trophoblastic cells, which are permissive to both InlA and InlB, even when using inlA-deficient strains. To extend the physiological relevance of LisCVs, it would be valuable to demonstrate their presence in vivo, such as in the placenta of wild-type mice, where InlA is dispensable, or in the liver, where LisCVs were previously observed in hepatocytes (Kortebi et al., 2017). In vivo evidence of LisCV formation and modulation would substantially enhance the study’s impact by underscoring its significance in Lm persistence and pathogenesis.

Impaired folate metabolism and ActA deregulation are not necessary on the same stream. Is it possible to add back FolP downstream metabolites, such as folic acid shown in Fig 5J, to address the role of folate metabolism on the ActA deregulation?

Reviewer #2: -Fig1D. Are these bacteria in LisCV or eSLAPs? Is there a marker the authors can use to test this? This question does not take away my enthusiasm for these studies of Listeria persistence in vacuoles. But it is important the authors use the two terms accurately.

-There are many steps in the Listeria life cycle, and also in the formation of LisCVs. It would help the reader to include a schematic showing these steps early in the paper (e.g. Figure 1). This would help describing the different assays and where they fit into the paradigm of LisCV formation. It must also be borne in mind that changes to metabolism and virulence gene expression could impact any/all of the steps that lead up to LisCV formation.

-Fig1D. Where are the images for V1? From Fig1E,F it seems there are very few bacteria, but where are they?

-The authors should consider adding a schematic summarizing all the strains and mutations they identified linked to loss of Listeria persistence.

**Part III – Minor Issues: Editorial and Data Presentation Modifications**

Reviewer #1: Fig. 1C. The change of intracellular bacterial load can be modulated by cell viability. Is there any information regarding the viability of the cells after 72 hours of infection? Did any strain kill the cells efficiently but show comparable intracellular bacterial load?

Caco2 is a human epithelial cell line used to identify inlA function. Are LisCVs observed in the cells?

L76. Both InlA and InlB are internalin family proteins. However, internalin specifically indicates InlA, InlB is not called internalin B. Better to say “internalin proteins InlA and InlB” here.

Fig. 1D. From the text we know the images were from cells infected by Lm for 72 hours. It is better to mention the information in the figure and/or the legends to help the readers.

L212-213. I am not sure if the results support the conclusion that “InlA-Δ isolates are a better source to find genes involved in intracellular infection.” This could be due to other factors in the genetic background of CC9 since all the identified strain are all of CC9 but none from CC121.

L214 and fig 2C. As shown in Fig 1D and 2C, Lm can be LAMP1 associated and polymerize actin in the cytosol in one single infected cell. Can the authors provide the proportion of LAMP1-colocalized Lm cells so that the contribution of lysosomal Lm can be highlighted? Location of the inset images in the larger field images shall be indicated.

L217 and L218. I assume the authors wanted to highlighted the identification of isolates in nature with altered intracellular persistence. However, we already know that non-hemolytic isolates deficient in LLO function are all defective in intracellular infection. The claim shall be refined.

Fig. 3ABC. LisCV size of actA mutant has not been quantified. Would be better to have it here.

L233-234 and S3 fig. The data do not support the description. Instead, the result shows reduced infection, which can be due to reduced dissemination or initial invasion. Time-lapse recording is needed to confirm that V2 remain confined to the initially infected cells.

L259-262. Any evidence for PrfA activation defect in V2? Is expression of any other genes in PrfA regulon, in addition to actA, affected in V2?

L269. Less for uncountable nouns. Use “fewer” here.

L336-337. Including the growth curve in the supporting information is helpful.

S1A and S1B movies. Precise time recording information showing time axis in the video is needed.

L508-510. Different MOI was used for CC9, CC121 and the reference strains for screening. Was this the case for other experiments? Better to mention this in the result and figure legends for clarification.

Reviewer #2: -line 64, the paragraph seems to end after one sentence… typo?

-Fig1D. What timepoint is shown in the IF images (should be stated in figure legend).

-line 179, “must be due to the acquisition of one or more mutations”. Recommend changing to “might be due”.

-line 370 “Some bacteria can even adopt a dual life style”… is there a good example the authors can discuss? Importantly, what is known about the genetic and metabolic factors that govern which lifestyle is utilized?

PLOS authors have the option to publish the peer review history of their article (what does this mean? ). If published, this will include your full peer review and any attached files.). If published, this will include your full peer review and any attached files.

**Do you want your identity to be public for this peer review?** For information about this choice, including consent withdrawal, please see our For information about this choice, including consent withdrawal, please see our Privacy Policy ..

Reviewer #1: No

Reviewer #2: No

**Figure resubmission:**

**Reproducibility:**



---

## [Editor Report · Decision Letter 1]

26 Feb 2026

Dear Dr Milohanic,

We are pleased to inform you that your manuscript 'Large-scale phenotyping and comparative genomics reveal genetic features of Listeria persistence in epithelial cells' has been provisionally accepted for publication in PLOS Pathogens.

Best regards,

Alice Prince

Section Editor

PLOS Pathogens

Alice Prince

Section Editor

PLOS Pathogens

Sumita Bhaduri-McIntosh

Editor-in-Chief

PLOS Pathogens

orcid.org/0000-0003-2946-9497

Michael Malim

Editor-in-Chief

PLOS Pathogens

orcid.org/0000-0002-7699-2064

The revisions nicely address the major concerns of the reviewers. This is a strong manuscript and adds our growing appreciation of the intracellular life styles of important pathogens.
---

## [Editor Report · Acceptance letter]

Dear Dr Milohanic,

We are delighted to inform you that your manuscript, "Large-scale phenotyping and comparative genomics reveal genetic features of Listeria persistence in epithelial cells," has been formally accepted for publication in PLOS Pathogens.

Best regards,

Sumita Bhaduri-McIntosh

Editor-in-Chief

PLOS Pathogens

orcid.org/0000-0003-2946-9497

Michael Malim

Editor-in-Chief

PLOS Pathogens

orcid.org/0000-0002-7699-2064